# Analyzing Neural Style Representations for Unsupervised Clustering: Visual Art as a Testbed

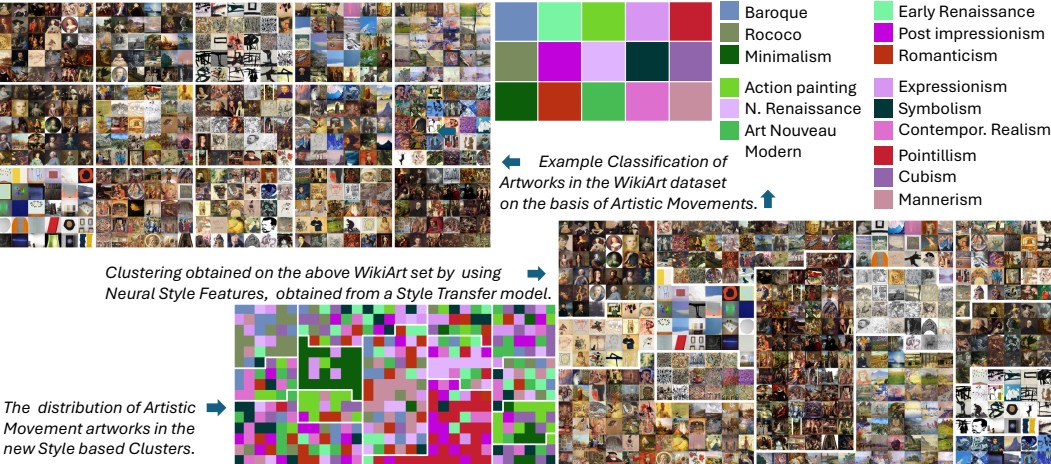

Figure 1: The figure shows a sample of the WikiArt dataset, which has ground truth clusters depicting various art movements from Baroque to Mannerism. The above artworks are re-clustered using the neural style representation $F_{StyleShot}$; the infographics show the distribution of ground-truth in different clusters. Even though the re-clustering through $F_{StyleShot}$ representation does not produce clusters that adhere to ground truth, we can see that the artworks present in the same cluster are similar to each other in terms of style, highlighting a fundamental discrepancy between historical art categorizations and perceptual style representations.

## Abstract

Neural networks claim to capture artistic style, but it remains unclear whether their representations can organize artworks in unsupervised settings, or which aspects of style they truly encode. We present the first comprehensive analysis of neural style representations for unsupervised clustering of visual artworks. Our study systematically compares representations derived from classification networks, generative models, diffusion architectures, and vision-language systems, including our novel language-based features. Using both real-world artwork collections and synthetically curated datasets, we evaluate how effectively these representations capture style across multiple definitions. Our results show that specialized style representations consistently outperform generic embeddings, yet no single representation works across all style definitions. This variability reflects the inherent ambiguity of "style" itself, revealing a gap between human perception, art-historical categories, and machine-learned features. Taken together, our findings position visual art as a rigorous testbed for advancing unsupervised representation learning with broader implications for digital curation, cultural heritage, and style-aware computer vision.

# 1 INTRODUCTION

**Style** is central to visual art. It embodies an artist's identity, emotional expression, cultural context, and aesthetic choices. In computer vision, style has also become an important dimension for understanding images beyond object categories. Neural representations of style now support a range of tasks such as style classification (Agarwal et al., 2015; Imran et al., 2023; Nunez-Garcia et al., 2022), style transfer (Gatys et al., 2016; Hong et al., 2023; Jing et al., 2019), and style-based retrieval (Ruta et al., 2023; 2021b). These tasks have driven the development of diverse architectures, from convolutional networks to generative models to vision-language systems, each claiming to capture artistic style effectively.

**Yet their effectiveness in unsupervised settings remains largely unknown.** Unsupervised analysis matters because it tests whether representations capture *intrinsic stylistic relationships* without labels - insights that supervised tasks cannot provide. It also has practical importance: large collections such as WikiArt or museum archives contain tens of thousands of artworks without consistent annotation. Curators and historians need computational tools that can automatically organize these collections, reveal latent stylistic patterns, and uncover artistic influences beyond fixed taxonomies (Refer Appendix A for a more detailed discussion.)

A deeper challenge is that "style" itself has **multiple valid definitions**. It may refer to broad art movements (e.g., Impressionism, Cubism), an individual artist's signature, low-level attributes such as color or brushstrokes, or even medium and technique (refer Section 4.2). Each definition yields a different but legitimate organization of artworks. Yet no prior work has systematically tested how neural representations behave across these alternative style definitions in unsupervised settings.

The few existing studies of unsupervised clustering of artworks rely on general-purpose image features, not style-specific ones (Castellano & Vessio, 2020). They fail to address the key issue of disentangling style from content and overlook the definitional complexity of artistic style. Moreover, the evaluation is hindered by the lack of controlled datasets that isolate style from content - a critical requirement for rigorously testing style-content disentanglement capabilities.

**This gap motivates our work.** We establish style-based clustering as a distinct computational paradigm and present the first comprehensive analysis of **16 state-of-the-art neural style representations** evaluated in the context of **unsupervised clustering across multiple definitions of style**.

This paper makes the following significant **contributions**:

**1. Comprehensive evaluation framework:** We develop a unified framework of datasets, clustering algorithms, and evaluation protocols for comparing how representations capture different notions of style.

**2. Controlled synthetic datasets:** We create synthetically curated datasets that systematically isolate style from content, enabling rigorous evaluation of style-content disentanglement - a capability that real-world datasets cannot provide.

**3. Benchmarking and analysis:** We provide the first systematic performance analysis across 16 neural representation methods and 2 clustering architectures, tested across multiple definitions of style using visual art as a testbed.

**4. Language style representations:** We introduce novel language-based style representations derived from captioning and concept annotation using vision-language models.

**Key Results.** Our study yields three main insights. First, style-specific representations, especially from style-transfer models, outperform generic embeddings in unsupervised clustering. Second, no single representation works across all definitions of style, reflecting both the ambiguity of "style" and the gap between perception, art-historical categories, and learned features. Third, clustering architectures enhance cluster geometry but not style fidelity, underscoring the tension between general unsupervised objectives and style-specific organization. These findings establish visual art as a rigorous testbed for advancing representation learning.

**Paper Organization.** Section 2 reviews related work; Section 3 outlines the style representations; Section 4 describes the experimental setup; Section 5 presents results and discussion; and Section 6 concludes with key takeaways and future directions.

## 2    RELATED WORK

Our work is the first systematic study of *style-based clustering* in visual art, comparing diverse representations of neural style in multiple definitions of style. We organize related work into three areas.

**Evaluation of Neural Style Representations:** Neural style representations have been evaluated, but almost exclusively in the context of downstream tasks. For supervised classification, Imran et al. (2023) tested CNN features such as VGG-19 and ResNet-50, while Chu & Wu (2018) analyzed Gram-based correlations. Varshney et al. (2023) explored fusion features for Madhubani art, and other works focused on artist-specific (Deng et al., 2020) or movement-based (El Vaigh et al., 2021) prediction. These evaluations show that neural features capture stylistic cues, but only relative to predefined categories.

In style transfer, representations have also been assessed, but in terms of synthesis quality. From Gram matrices (Gatys et al., 2016) to GAN, transformer and diffusion-based methods (Jing et al., 2019; Hong et al., 2023; Sohn et al., 2023; Yang et al., 2023), evaluation typically concerns visual fidelity or transfer control. While such models implicitly encode stylistic information, their capacity to support unsupervised clustering across different definitions of style has not been examined.

In contrast, we systematically evaluate a wide range of representations, including classification-based, transfer-based, and multimodal-based representations in the context of *unsupervised clustering of artworks*.

**Unsupervised Artwork Clustering:** Unsupervised clustering of artworks has been explored only in limited ways, and never with a focus on style. Early efforts relied on manual grouping (Spehr et al., 2009) or simple K-means on image features (Coates & Ng, 2012; Ng et al., 2001). Gairola et al. (2020) used Gram features for soft style assignments via K-means. More broadly, deep clustering has shown strong performance across domains such as segmentation, medical imaging, and anomaly detection (Yang et al., 2018; Chen et al., 2022; Kart et al., 2021; Ma et al., 2021; Adaloglou et al., 2023; Alkin et al., 2025). For art, Castellano and Vessio applied deep clustering to image and DenseNet features (Castellano & Vessio, 2020; 2022), but without addressing style-content disentanglement or comparing specialized style representations.

**Clustering Evaluation:** Clustering quality is typically assessed with both internal and external metrics. Internal measures include Silhouette Coefficient (Rousseeuw, 1987) and Calinski–Harabasz Index (Caliński & JA, 1974), among others Dunn (1973); Davies & Bouldin (2009); Glielmo et al. (2021); Golalipour et al. (2021); Guo et al. (2023); Zhang et al. (2021). We adopt Silhouette and Calinski–Harabasz as complementary internal metrics. External evaluation relies on clustering accuracy (Fränti & Sieranoja, 2024), Adjusted Rand Index (ARI) (Rand, 1971), and Normalized Mutual Information (NMI) (Shannon, 1948), which are widely used for artwork clustering (Yao et al., 2024; Castellano & Vessio, 2022; Zhong et al., 2021). Since clustering accuracy can be misleading under class imbalance, we emphasize ARI and NMI as more robust measures.

**Summary:** Prior work has developed many style representations and attempted clustering of artworks, but evaluations remain tied to supervised tasks or generic features. No study has systematically compared neural style representations for *unsupervised clustering across multiple definitions of style*-the central gap our work addresses.

## 3    STYLE REPRESENTATION EXPLORATION

The effectiveness of style-based clustering depends critically on the choice of neural representations. Different architectures trained for classification, transfer, or multimodal alignment may capture style in fundamentally different ways, yet their suitability for *unsupervised* organization remains unclear. To address this, we explore five categories of style representations (Fig. 3):

1. **Generic task-based representations** from broadly trained encoders (e.g., DenseNet, DINOv2, LongCLIP).

2. **Style feature-based representations** that compute explicit statistics such as Gram matrices or introspective style attribution.

Table 1: Summary of style representations explored. Our contributions are highlighted in **bold**.

| Category | Model | Source / Training | Representation Extracted |
|---|---|---|---|
| Generic Task-based | DenseNet ($F_{Dense}$) | ImageNet CNN (Huang et al., 2018) | 1024-dim pooled final layer features |
| | Vision–Language ($F_{LongCLIP}$) | LongCLIP (Zhang et al., 2024) | Joint image–text embeddings (long captions) |
| | Self-supervised ($F_{DINO}$) | DINOv2 (Oquab et al., 2023) | SSL-based semantic embeddings |
| Style Feature-based | Gram Matrices ($F_{Gram}, F_{g \cdot c}$) | VGG19 conv5_1 (Gatys et al., 2015) | Gram matrix statistics; cosine similarity variant |
| | Introspective Style Attribution ($F_{IntroStyle}$) | Diffusion UNet (Kumar et al., 2025) | Channel-wise mean/variance embeddings |
| Style-Transfer | StyleGAN ($F_{StyleGAN}$) | GAN (Karras et al., 2019) | Latent $w$ vector |
| | Stytr$^2$ ($F_{Stytr2}$) | Transformer (Deng et al., 2022) | Encoder outputs from patch embeddings |
| | StyleShot ($F_{StyleShot}$) | Transformer (Gao et al., 2024) | Multi-scale patch embeddings |
| | Mamba-ST ($F_{Mamba}$) | VSSM (Botti et al., 2024) | Style encoder representations |
| | DEADiff ($F_{DEADiff}$) | Diffusion T2I (Qi et al., 2024) | Q-Former embeddings (style disentangled) |
| **Language-based (ours)** | Style Caption ($F_{StyleCap}$) | InternVL2 + LongCLIP | Encoded style captions |
| | Style Concept Annotations ($F_{Annot}$) | InternVL2 + taxonomy | Structured annotations across 59 concepts |
| Style-Trained | Contrastive Style Descriptors ($F_{CSD}$) | ViT on LAION Aesthetics (Somepalli et al., 2024) | Contrastive embeddings with style tags |
| | **Artwork-Trained ViTs (ours)** | Fine-tuned on WikiArt (Tan et al., 2016) | (i) Art movement ($F_{ArtMove}$), (ii) Artist ($F_{Artist}$) |

3. **Style-transfer representations** from GANs, transformers, and diffusion models developed for synthesis and transfer tasks.

4. **Language-based representations (ours)** that provide interpretable embeddings derived from captions and structured concept annotations.

5. **Style-trained image models**, including contrastively trained descriptors and **artwork-trained ViTs (ours)** fine-tuned on WikiArt.

Table 1 summarizes all representations considered, while full details - including the implementation of our novel languag-based and artwork-trained models - are provided in Appendix B.

## 4  EXPERIMENTAL SETUP AND EVALUATION

We design our experimental setup to test how different neural representations capture style across multiple definitions of the term. This requires (i) clustering models that span simple and expressive approaches, (ii) datasets that operationalize different notions of style, and (iii) evaluation metrics that combine algorithmic rigor with human judgment.

### 4.1  CLUSTERING MODELS

To assess clustering effectiveness, we use two complementary approaches:

**K-Means Clustering**: A standard baseline that partitions the feature space by Euclidean distance MacQueen et al. (1967). It provides a simple test of whether style information is directly encoded in representations (Appx. Fig. 6A).

**Deep Embedded Clustering (DEC)**: A stronger model that jointly optimizes embeddings and cluster assignments via an autoencoder Xie et al. (2016). DEC captures non-linear relationships and

tests whether style emerges only when representations are refined through deep clustering (Appx. Fig. 6B).

## 4.2 DATASETS

A major challenge in studying style-based clustering is the lack of ground-truth datasets, combined with the fact that *style in artworks* admits multiple valid definitions. In the literature, style has been treated as: statistical properties of neural representations (Gatys et al., 2015); distinctive signatures of artists or movements (Elgammal et al., 2017); low-level attributes such as color, texture, or brushstrokes (Liu et al., 2024); domain-specific distributions (Zhu et al., 2020); perceptually significant cues from human studies (Muller, 1979); and transformative operations applied to content (Huang et al., 2025).

To capture these diverse perspectives, we employ four complementary datasets: (i) **WikiArt-ArtMove** for movement-based categories; (ii) **WikiArt-Artist** for individual artistic signatures; (iii) **DomainNet-3k** for disentangling style (medium) from content (object category); and (iv) **Synthetic Curated Datasets (MSC/MMC)**, **introduced in this work**, which provide controlled style–content separation via style transfer.

A full summary of dataset statistics, style definitions, and usage purposes is provided in Table 2, with further details and sample artworks in Appendix E.

Table 2: Datasets used in our study, each operationalizing a different definition of style for unsupervised clustering evaluation. **Our synthetic curated datasets (MSC/MMC) are novel contributions** providing controlled style–content disentanglement.

| Dataset | Images | Clusters | Definition & Purpose | Notes / Examples |
|---|---|---|---|---|
| **WikiArt-ArtMove** | 78,978 | 27 | Art movements (e.g., Realism, Baroque, Impressionism). Tests alignment with art-historical categories. | Large-scale benchmark covering 27 movements across centuries. Provides ground-truth groupings for movement-based evaluation. See Fig. 8(a); Figs. 14–18. |
| **WikiArt-Artist** | 25,550 | 40 | Individual artist signatures (e.g., Picasso, Van Gogh, Dali). Evaluates artist-specific style alignment. | Subset of 40 artists with distinctive personal styles. Useful for testing whether models capture fine-grained stylistic variation. See Fig. 8(a); Figs. 19–23. |
| **DomainNet-3k** | 3,000 | 6 style + 50 content | Medium-based domains (clipart, sketch, photo, painting, etc.). Used to assess content–style disentanglement. | Cross-domain dataset where the same objects appear in different styles. Tests whether representations separate style from semantics. See Fig. 8(b); Fig. 12. |
| **Synthetic Curated (MSC/MMC)** | 4,000 | 40 style + 100 content | Style-transfer images (color, texture, shading). **Introduced in this work for controlled disentanglement experiments.** | Generated via StyleShot and Mamba-ST using references from Munch, WikiArt, Brueghel, and Clipart. Provides synthetic but well-controlled style–content separations. See Fig. 8(c)–(d), 10, 24–28. |

## 4.3 QUANTITATIVE EVALUATION METRICS

Our evaluation framework employs four complementary metrics - divided into internal and external measures - to comprehensively assess clustering quality. Internal metrics measure inherent properties (cohesiveness and separation) of the clustering itself, while external metrics use ground truth-style labels to measure alignment with predefined categories. See Appendix D for metric details.

### 4.3.1 INTERNAL EVALUATION METRICS

**Silhouette Coefficient (SC)** Rousseeuw (1987) quantifies how appropriately each artwork is assigned to its cluster by measuring the ratio between intra-cluster cohesion and inter-cluster separation. Values range from -1 to +1, with higher values indicating more distinct cluster boundaries. By

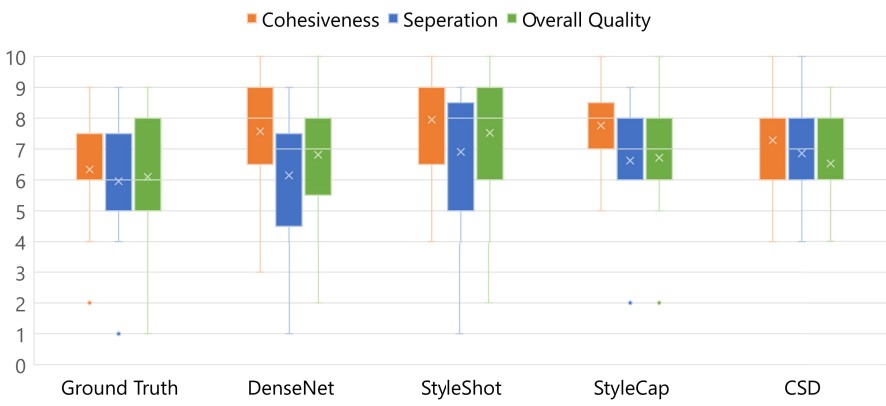

Figure 2: Box plots for the survey conducted on the clusters obtained from 450 samples from the *WikiArt-ArtMove* dataset. Clusters were obtained on the subset using $F_{Dense}$, $F_{StyleCap}$, $F_{StyleShot}$ and $F_{CSD}$ features and K-Means clustering model. The survey was conducted on the art movement ground truth as well. The survey included questions relating to cluster cohesion, cluster separation, and overall clustering quality. Overall, 25 participants responded to the survey.

evaluating individual data point placement, SC reveals how effectively each representation distinguishes the unique attributes of artworks and separates them from stylistically different works.

**Calinski-Harabasz Index (CHI)** Caliński & JA (1974) captures clustering quality at both individual and cluster levels by computing the ratio of between-cluster dispersion to within-cluster dispersion. Higher values indicate more compact and well-separated clusters. This multi-level perspective helps evaluate each representation's capacity to differentiate stylistic attributes across diverse artworks, providing insight into both local and global separation efficacy.

### 4.3.2 EXTERNAL EVALUATION METRICS

**Adjusted Rand Index (ARI)** Rand (1971) measures agreement between predicted clusters and ground truth style categories, with values ranging from -1 to 1. Scores above 0 indicate correspondence beyond random chance, with 1 representing perfect alignment. By quantifying the agreement between algorithmically derived clusters and style-definition-based categories, ARI directly assesses how well a representation captures the particular style definition embodied in the ground truth.

**Normalized Mutual Information (NMI)** Vinh et al. (2010) quantifies the statistical dependency between predicted clusters and ground truth categorizations. Ranging from 0 to 1, higher values indicate a stronger correlation between the information contained in both clusterings. NMI effectively measures how much knowing an artwork's algorithmic cluster reduces uncertainty about its ground truth style category, providing a complementary perspective to ARI.

We can apply internal metrics (SC, CHI) across all experiments as they require no ground truth, while external metrics (ARI, NMI) are used exclusively with datasets where definitive style labels are available.

## 4.4 HUMAN PERCEPTUAL EVALUATION

**Qualitative Comparison:** We illustrate clustering outcomes by sampling artworks from the ground-truth datasets and visualizing the clusters produced by different representations. While limited in scope, these examples serve as an accessible reference for assessing the effectiveness of each clustering method (see Appendix G).

**Survey**: Finally, because style is inherently subjective, we complement algorithmic metrics with human evaluation. Twenty-five participants with art backgrounds rated clustering results on *cohesiveness, separation, and overall quality* (1–10 scale). Importantly, no style definition was imposed—participants applied their own aesthetic criteria. This lets us capture perceptual judgments of style that cannot be reduced to categorical ground truth (see Appendix F).

| Representations | | ARI | | NMI | | | Representations | | ARI | | NMI | |
|---|---|---|---|---|---|---|---|---|---|---|---|---|
| | | K-Means | DEC | K-Means | DEC | | | | K-Means | DEC | K-Means | DEC |
| $F_{Generic}$ | $F_{Dense}$ | 0.052 | 0.059 | 0.172 | 0.177 | | $F_{Generic}$ | $F_{Dense}$ | 0.0001 | 0.0001 | 0.007 | 0.007 |
| | $F_{LongCLIP}$ | 0.099 | 0.078 | 0.288 | 0.271 | | | $F_{LongCLIP}$ | 0.076 | 0.053 | 0.307 | 0.271 |
| | $F_{DINO}$ | 0.052 | 0.043 | 0.189 | 0.176 | | | $F_{DINO}$ | 0.0976 | 0.0813 | 0.29 | 0.267 |
| $F_{StyleFeat}$ | $F_{Gram}$ | 0.042 | 0.064 | 0.13 | 0.151 | | $F_{StyleFeat}$ | $F_{Gram}$ | 0.0003 | 0.0002 | 0.009 | 0.008 |
| | $F_{g.c}$ | 0.012 | 0.012 | 0.034 | 0.034 | | | $F_{g.c}$ | 0.0002 | 0.0001 | 0.008 | 0.008 |
| | $F_{IntroStyle}$ | 0.04 | 0.03 | 0.15 | 0.124 | | | $F_{IntroStyle}$ | 0.02 | 0.01 | 0.2 | 0.179 |
| $F_{ST}$ | $F_{StyleGAN}$ | 0.034 | 0.021 | 0.103 | 0.095 | | $F_{ST}$ | $F_{StyleGAN}$ | 0.069 | 0.067 | 0.214 | 0.204 |
| | $F_{Stytr2}$ | 0.021 | 0.03 | 0.078 | 0.092 | | | $F_{Stytr2}$ | 0.0017 | 0.0002 | 0.008 | 0.007 |
| | $F_{Mamba}$ | 0.034 | 0.029 | 0.117 | 0.085 | | | $F_{Mamba}$ | 0.075 | 0.095 | 0.239 | 0.229 |
| | $F_{StyleShot}$ | 0.055 | 0.043 | 0.161 | 0.151 | | | $F_{StyleShot}$ | 0.0005 | 0.0009 | 0.008 | 0.008 |
| | $F_{DEADiff}$ | 0.1 | 0.081 | 0.3 | 0.273 | | | $F_{DEADiff}$ | 0.22 | 0.192 | 0.45 | 0.413 |
| $F_{Lang}$ | $F_{StyleCap}$ | 0.071 | 0.058 | 0.225 | 0.192 | | $F_{Lang}$ | $F_{StyleCap}$ | 0.0008 | 0.006 | 0.01 | 0.014 |
| | $F_{Annot}$ | 0.069 | 0.043 | 0.208 | 0.162 | | | $F_{Annot}$ | 0.0006 | 0.052 | 0.019 | 0.197 |
| $F_{Train}$ | $F_{CSD}$ | 0.12 | 0.095 | 0.33 | 0.232 | | $F_{Train}$ | $F_{CSD}$ | 0.31 | 0.262 | 0.51 | 0.461 |
| | $F_{ArtMove}$ | 0.27 | 0.161 | 0.45 | 0.325 | | | $F_{Artist}$ | 0.54 | 0.471 | 0.68 | 0.634 |

Table 3: Quantitative evaluation on the **WikiArt-ArtMove (left)** dataset and the **WikiArt-Artist (right)** dataset for both K-Means and DEC model with their respective ground truths. The  best ,  second best ,  third best , and the  worst  results are highlighted for each metric. The WikiArt-Artist dataset of 40 artists contains 25,550 artworks from the artists with the highest amount of artworks. The range of values for each evaluation metric is: *ARI*: -1 to 1, *NMI*: 0 to 1. For qualitative comparison of the WikiArt-ArtMove refer to Appendix Figures 14, 15, 16, 17, and 18. For qualitative comparison of WikiArt-Artist refer to Appendix Figure 19, 20, 21, 22, and 23.

## 5 RESULTS AND DISCUSSION

Our work investigates several fundamental questions at the intersection of neural representations and artistic style clustering. Specifically, we ask: (i) Does style-based clustering require specialized neural representations beyond general-purpose image features? (ii) How do different neural style representations perform in capturing stylistic similarities across clustering tasks? (iii) What influence do clustering architectures have on performance outcomes and are certain algorithms inherently better suited to style-based organization? (iv) Can a single representation adequately capture diverse definitions of style across contexts, or are specialized representations required for different stylistic dimensions? (v) Do artistic styles exhibit structural or hierarchical relationships that clustering can uncover?

We address these questions through a systematic evaluation of 16 distinct neural style representations and two clustering approaches (K-Means and Deep Embedded Clustering, DEC). Performance is measured along two complementary axes: (a) style-specific clustering quality (Adjusted Rand Index and Normalized Mutual Information) and (b) general cluster structure (Silhouette Coefficient and Calinski-Harabasz Index). Our findings are organized around five research questions, corresponding to the subsections below.

### 5.1 DO STYLE-BASED CLUSTERING TASKS REQUIRE SPECIALIZED REPRESENTATIONS?

**Finding 1:** Generic image representations fail to capture the nuanced stylistic elements of visual artworks.

On the DomainNet dataset (refer to Table 8 and Figures 12, 13 in the Appendix), generic representations trained for broad image tasks ($F_{Dense}$, $F_{LongCLIP}$, $F_{DINO}$) were unable to disentangle content and style (domain) and performed poorly across both evaluation dimensions. In contrast, specialized style representations ($F_{StyleCap}$, $F_{CSD}$, $F_{Mamba}$) demonstrated much stronger performance in style-based clustering. While some style-specific representations (e.g., $F_{StyleCap}$, $F_{CSD}$) retained partial sensitivity to content, others like $F_{Mamba}$ specialized narrowly on style. This establishes the need for explicitly style-focused features in artwork clustering. Refer to Appendix G.1 for more details on the experiment.

### 5.2 HOW DO NEURAL STYLE REPRESENTATIONS PERFORM UNDER TRADITIONAL ART-HISTORICAL DEFINITIONS OF STYLE?

**Finding 2:** All neural style representations perform poorly under traditional art-historical style definitions.

| Features | | ARI | | NMI | | SC | | | CHI | | |
|---|---|---|---|---|---|---|---|---|---|---|---|
| | | K-Means | DEC | K-Means | DEC | Base | K-Means | DEC | Base | K-Means | DEC |
| $F_{Generic}$ | $F_{Dense}$ | 0.594 | 0.611 | 0.893 | 0.861 | 0.078 | 0.071 | 0.709 | 41.89 | 41.06 | 3379.81 |
| | $F_{LongCLIP}$ | 0.541 | 0.471 | 0.653 | 0.592 | 0.047 | 0.031 | 0.641 | 53.41 | 51.67 | 4431.23 |
| | $F_{DINO}$ | 0.371 | 0.334 | 0.582 | 0.489 | 0.225 | 0.221 | 0.413 | 261.34 | 231.85 | 2347.51 |
| $F_{StyleFeat}$ | $F_{Gram}$ | 0.837 | 0.684 | 0.932 | 0.889 | 0.221 | 0.205 | 0.47 | 270.27 | 293.25 | 1380.32 |
| | $F_{g.c}$ | 0.078 | 0.079 | 0.343 | 0.344 | -0.229 | 0.28 | 0.51 | 805.3 | 25425.8 | 84313.98 |
| | $F_{IntroStyle}$ | 0.71 | 0.643 | 0.84 | 0.793 | 0.113 | 0.11 | 0.401 | 131.24 | 117 | 1731.38 |
| $F_{ST}$ | $F_{StyleGAN}$ | 0.5 | 0.478 | 0.758 | 0.719 | -0.03 | -0.003 | 0.869 | 17.65 | 19.24 | 12064.25 |
| | $F_{Stytr2}$ | 0.91 | 0.676 | 0.95 | 0.867 | 0.482 | 0.45 | 0.377 | 1065.08 | 1132 | 3019.4 |
| | $F_{Mamba}$ | 0.91 | 0.771 | 0.96 | 0.919 | 0.443 | 0.42 | 0.526 | 652.4 | 646.91 | 7759.72 |
| | *$F_{StyleShot}$ | 0.9 | 0.87 | 0.97 | 0.951 | 0.399 | 0.39 | 0.696 | 364.31 | 364.6 | 4846.23 |
| | $F_{DEADiff}$ | 0.84 | 0.812 | 0.91 | 0.898 | 0.06 | 0.04 | 0.523 | 31.73 | 28.53 | 7341.23 |
| $F_{Lang}$ | $F_{StyleCap}$ | 0.347 | 0.334 | 0.565 | 0.567 | 0.018 | 0.04 | 0.827 | 30.99 | 39.09 | 59267.93 |
| | $F_{Annot}$ | 0.213 | 0.214 | 0.467 | 0.457 | -0.0003 | 0.027 | 0.936 | 31.535 | 44.25 | 57340.58 |
| $F_{Train}$ | $F_{CSD}$ | 0.96 | 0.831 | 0.98 | 0.922 | 0.281 | 0.27 | 0.502 | 195.82 | 192.79 | 1858.74 |
| | $F_{Artist}$ | 0.0002 | 0.0001 | 0.015 | 0.012 | 0.12 | 0.11 | 0.14 | 87.39 | 77.57 | 121.33 |
| | $F_{ArtMove}$ | 0.0003 | 0.0002 | 0.015 | 0.013 | 0.15 | 0.13 | 0.15 | 93.41 | 88.49 | 98.63 |

Table 4: Metrics scores for the **Mixed curated** dataset created using **StyleShot** (MSC) for all features for both K-Means and DEC model. The best , second best , third best , and the worst results are highlighted for each metric. MSC contains 4000 images and 40 different styles. The range of values for each metric is: *ARI*: -1 to 1, *NMI*: 0 to 1, *SC*: -1 to 1, and *CHI*: 0 to $\infty$. The **Base** column indicates the SC and CHI values with perfect ground truth and no modification to the input embedding. *As $StyleShot$ was used to create the curated dataset, we excluded $F_{StyleShot}$ from ranking. For qualitative comparison, refer to Appendix Figures 24, 25, 26, 27, and 28.

On WikiArt's art movement and artist labels, all models achieved low clustering scores (Table 3). Even though artist-trained ($F_{Artist}$) and movement-trained ($F_{ArtMove}$) features performed relatively better on their respective definitions, overall performance remained low, reflecting the complexity of WikiArt's style categories. Language-based features ($F_{Lang}$) showed modest advantages over visual descriptors ($F_{Gram}$, $F_{ST}$).

Human evaluation (Figure 2) further revealed a disconnect between formal art-historical categories and intuitive human perception: participants often did not group artworks according to WikiArt's style labels. For further details on our human evaluation survey, please refer to Appendix F.

### 5.3 WHICH REPRESENTATIONS BEST CAPTURE PERCEPTUAL STYLE FEATURES?

**Finding 3:** Neural style transfer architectures yield the most effective representations for clustering when style is defined via perceptual features such as texture, color, or line quality.

On our synthetically curated datasets (Table 4 and Appendix Table 9), representations from style transfer models ($F_{ST}$) consistently achieved top performance, outperforming both traditional Gram-based features and newer diffusion-based descriptors. Contrastive style descriptors ($F_{CSD}$) also performed strongly. Diffusion-based style features ($F_{DEADiff}$, $F_{IntroStyle}$) excelled on curated datasets but underperformed on WikiArt, suggesting they capture simple perceptual traits better than historically grounded artistic styles. Language-based features ($F_{Lang}$) showed moderate style-capturing ability but excelled in internal cluster quality (SC, CHI), indicating semantic interpretability rather than purely visual style similarity.

### 5.4 HOW DO CLUSTERING ARCHITECTURES INFLUENCE STYLE-BASED ORGANIZATION?

**Finding 4:** K-Means achieves slightly higher style accuracy, while DEC produces more distinct clusters but does not necessarily improve style alignment.

K-Means consistently outperformed DEC on style-specific metrics (ARI, NMI). However, DEC produced higher SC and CHI scores, revealing better geometric separation in latent space. Importantly, these gains in cluster geometry did not always translate into better style-based clustering (Refer to Appendix Figure 7 in Appendix C for quantitative graphical comparison). DEC also offered computational efficiency, reducing runtime substantially on large datasets.

### 5.5 DO NEURAL REPRESENTATIONS REVEAL HIERARCHICAL RELATIONSHIPS AMONG ARTISTIC STYLES?

**Finding 5:** Neural style representations expose intricate hierarchical structures in artistic style corpora.

Two lines of evidence support this : (i) *Cluster Distribution Analysis*: models like $F_{Gram}$ grouped artworks into a few dominant clusters, which revealed fine-grained subclusters upon further partitioning; (ii) *Hierarchical Clustering*: dendrograms built on WikiArt using $F_{StyleCap}$ demonstrated coherent multi-level style organizations.(Refer to Appendix H for details. Together, these findings indicate that artistic style is inherently hierarchical, and clustering approaches should account for this rather than assuming flat structures.

### 5.6 LIMITATIONS

Our study has three main limitations: **(i) Ground Truth Definitions:** We evaluate clustering against three style definitions (synthetic datasets, WikiArt categories, and DomainNet domains). Broader definitions of style remain unexplored. **(ii) Visual Artworks Domain:** Our analysis focuses on artworks, but the framework could generalize to domains like fashion, music, or architecture. **(iii) LLM's Ability to Interpret Styles for Language Representations:** The quality of our language-based features depends on how well vision-language models interpret artistic style. While presently imperfect, these representations establish a useful benchmark for future multimodal style studies.

### SUMMARY OF FINDINGS

Overall, style-based clustering is highly sensitive to both the choice of representation and the definition of style. Specialized features outperform generic embeddings, but no single representation works consistently across all style definitions, reflecting the inherent ambiguity of "style." Clustering architectures introduce additional trade-offs: improved geometry does not guarantee improved style fidelity. Finally, style-based clustering uncovers hierarchical structures that future methods should explicitly model. These results position visual art as a demanding but valuable testbed for advancing unsupervised representation learning.

## 6 CONCLUSION

We established style-based clustering of visual artworks as a distinct computational task, highlighting its methodological challenges and practical significance. Our framework systematically evaluated sixteen neural style representations—spanning classification networks, generative models, diffusion-based architectures, and our novel language-based features—across multiple style definitions and clustering settings.

Our experiments reveal five key insights: (1) specialized style features outperform generic embeddings, (2) all models struggle under traditional art-historical definitions, (3) style transfer models excel when style is defined via perceptual attributes, (4) clustering architectures trade off between cluster geometry and style fidelity, and (5) style organization is inherently hierarchical. These findings demonstrate both the promise and the limitations of current neural style representations for unsupervised clustering.

Beyond visual art, the implications are broader. Many domains - from fashion and design to architecture and music - face similar challenges where domain-specific style or aesthetic choices must be disentangled from underlying semantic content. Visual art provides a demanding testbed for probing these issues, exposing the limits of current models and guiding the design of future representations that capture nuanced, multi-faceted notions of style.

In short, neural style representations remain powerful but imperfect tools for unsupervised organization. Our analysis not only clarifies their strengths and weaknesses but also points toward opportunities for building representations that bridge human perception, cultural constructs, and machine learning.

**Key Takeaway:** Specialized neural style representations outperform generic embeddings for clustering visual art, but no representation works consistently across all style definitions. This highlights the inherent ambiguity of "style" and positions visual art as a rigorous testbed for advancing unsupervised representation learning.

## 7 DECLARATION OF LLM USAGE

LLM (or Grammarly) has been used in the paper only for writing, editing, or formatting purposes and does not impact the core methodology. In our experimental work, we have used Large Vision Language Model (LVLM) (LLMs with a Vision component) for implementing an original method of obtaining new style representations for artworks by annotating the styles present in the artworks with the LVLM and encoding the textual style descriptions. This was the core requirement of our work.

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

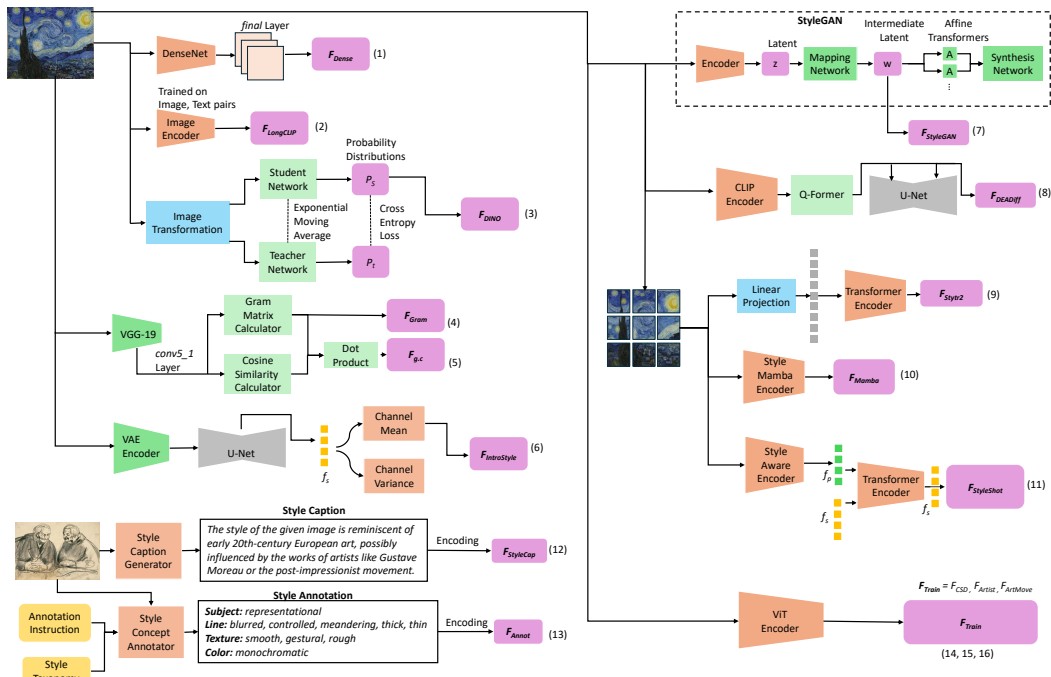

Figure 3: The sub-figures 1-16 show the Architectures for extracting various neural style representations where representation 1-3 are Generic Task-based Models, 4-6 are from Style Feature-based Models, 7-11 are from Style Transfer based Models, 12-13 are from Language models and 14-16 are from Style Trained models.

## A  STYLE-BASED CLUSTERING AND ITS SIGNIFICANCE

Artistic style becomes clear when we see multiple works together, not from single pieces alone. Art experts have traditionally done this pattern recognition by hand. Some researchers have tried using crowdsourcing for more detailed style categories (Ruta et al., 2021a), but this approach faces challenges: style is subjective, and finding patterns in large collections requires enormous human effort. Computational methods can transform how we analyze style at scale.

Computer-assisted style clustering offers several key advantages. First, it can analyze collections far larger than any human could handle, finding hidden relationships that might never be discovered otherwise. Second, it can identify subtle style categories that go beyond traditional art history labels. Artists often work in many different styles throughout their careers - much more varied than broad terms like "Cubism" or "Impressionism" can capture.

Third, computational clustering can trace how artistic styles evolve over time by mapping relationships between works. This creates new stories about how art developed within individual careers and across broader movements. Fourth, these groupings create valuable resources for style transfer applications, from creative tools to museum exhibits. Finally, style-based clusters help in education by letting students explore how styles connect and vary through organized groups rather than random examples.

These benefits show that style-based clustering is more than just a technical tool - it's a valuable method for art research, digital curation, and helping people engage with visual culture.

## B  NEURAL STYLE REPRESENTATIONS DETAILS

The essential first component in style-based clustering is the choice of neural representations that can help us identify the style in an artwork. In our main paper, we briefly touched upon the different

representations utilized in our evaluation framework. In this section, we provide additional details for each of the representations used. In general, we explore five categories of style representations (Fig. 3):

1. **Generic task-based representations** from broadly trained encoders (e.g., DenseNet, DINOv2, LongCLIP).

2. **Style feature-based representations** that compute explicit statistics such as Gram matrices or introspective style attribution.

3. **Style-transfer representations** from GANs, transformers, and diffusion models developed for synthesis and transfer tasks.

4. **Language-based representations (ours)** that provide interpretable embeddings derived from captions and structured concept annotations.

5. **Style-trained image models**, including contrastively trained descriptors and **artwork-trained ViTs (ours)** fine-tuned on WikiArt.

## B.1 GENERIC TASK BASED REPRESENTATIONS ($F_{Generic}$)

In the field of machine learning, popular image-based representations made for specific tasks were found to perform well in various computer vision tasks. In our experiments, we test three different kinds of representations found to perform well for different vision-based tasks, in the context of style-based clustering.

### B.1.1 DENSENET REPRESENTATIONS ($F_{Dense}$)

Some previous approaches for clustering visual artworks (Castellano & Vessio, 2022) utilize the last layer of DenseNet (Huang et al., 2018) to extract representations from the artworks and incorporate them into the DEC model. The last layer would contain rich information about the artwork, as each layer of the DenseNet is connected to all its previous layers. In our exploration, we utilize $F_{Dense}$ representations to check whether these representations can also be effective in style-based clustering for visual artworks. Similar to Castellano & Vessio (2022), after obtaining the representations for each artwork in the dimensions $1024 \times 7 \times 7$, we apply global average pooling (GAP) (Lin et al., 2014) to obtain 1024-representation vector for each artwork.

### B.1.2 VISION LANGUAGE REPRESENTATION ($F_{LongCLIP}$)

Vision language representations are one of the most popular categories of representations as they are utilized for multiple image-based downstream tasks such as image classification, image captioning, image generation, etc. These models utilize text-image pairs to learn a common representation between the text and the image associated with it. For the task of style-based clustering, we test one of the more recent CLIP (Radford et al., 2021) models, LongCLIP (Zhang et al., 2024). Similar to the CLIP model, LongCLIP utilizes a text encoder and an image encoder to learn the common representations between a text and image pair. LongCLIP improves upon CLIP by introducing two fine-tuning strategies, knowledge preserved stretching of positional embedding and primary component matching of CLIP representations. This strategy allows LongCLIP to support longer text provide better representations for different downstream tasks. We test these representations for style-based clustering.

### B.1.3 SELF-SUPERVISED REPRESENTATION ($F_{DINO}$)

The representations from self-supervised Vision Transformer Models (ViTs) were found to have emerging properties which stand out when compared to convolutional neural networks (Caron et al., 2021). Initially utilized for semantic segmentation, the DINO model (Caron et al., 2021) has also been used for different downstream tasks such as Image classification, Image restoration, etc. For our work, we utilize the DINOv2 model (Oquab et al., 2023). DINOv2 utilizes vision transformers as the student-teacher network where two different transformations of an input image are passed to each network. Both networks follow the same architecture while the parameters vary. The student network learns to predict the global representations of an input image by minimizing the cross entropy loss between the student and teacher representation.

## B.2 STYLE FEATURE-BASED STYLE REPRESENTATIONS ($F_{StyleFeat}$)

In the early adoption of neural networks, Gatys et al. (2015) found that the gram matrices of representations extracted from the intermediate layers of the VGG network contain information about the style features associated with a certain input image. Following this work, many different works explored the extraction of intermediate representations from different models, as they believed the intermediate representations would contain style features present in the input image. We utilize such representations from these works for evaluating their efficacy in style-based clustering.

### B.2.1 GRAM MATRICES BASED STYLE REPRESENTATIONS ($F_{Gram}$ AND $F_{g \cdot c}$)

Gram based representations were introduced by Gatys et al. (2015) for addressing the style transfer problem. Chu & Wu (2016; 2018) explore different combinations of representations extracted from different layers of CNN, as well as different mathematical correlations and combinations of these representations for style classification. They observe that the gram matrices of representations obtained from $conv5\_1$ ($F_{Gram}$) yield one of the best results in the classification task. They also observe that the dot product of $F_{Gram}$ and cosine similarity of the representations extracted from $conv5\_1$ ($F_{g \cdot c}$) also produces good results. In our work, we use $F_{Gram}$ and $F_{g \cdot c}$ and explore their impact on unsupervised style-based clustering. We reduce the dimensions of $F_{Gram}$ and $F_{g \cdot c}$ from $512 \times 512$ to $512$ for each artwork using GAP.

### B.2.2 INTROSPECTIVE STYLE ATTRIBUTION REPRESENTATION ($F_{IntroStyle}$)

We extract the style features from the IntroStyle model (Kumar et al., 2025), a diffusion based style attribution model. It finds the attribution by computing the style features from images and matching them based on Wasserstein distance metric. Following the standard process, an image is initially fed into the VAE encoder and the obtained latent is noised to timestep t. The noised latent is passed through the denoising network and the features are extracted from the intermediate layers of the unet. These features are known to be effective at disentangling image properties proving helpful for several downstream tasks. From these intermediate features, the feature statistics are computed as channel-wise means and variances and used as compact style representations. We extract these representations and use them as $F_{IntroStyle}$ in our work.

## B.3 OUR CONTRIBUTION: LANGUAGE STYLE REPRESENTATIONS ($F_{Lang}$)

In this section, we discuss in detail our two proposed approaches for style-based clustering that explicitly factor in the artistic style information from the artworks through visio-language-based models. The style representations presented thus far capture the style of an artwork in an interpretation which is difficult for a human to interpret in a meaningful way. Providing the style with some sort of interpretability that is easily understood by a human could provide a meaningful way to study the style aspects of an artwork. Expressing the style of an artwork through a textual medium allows a user to easily interpret the style of an artwork and find the correlation between the clustered artworks. To this end, we propose two types of textual style representations ($F_{Lang}$): $F_{StyleCap}$ and $F_{Annot}$.

### B.3.1 ARTWORK STYLE CAPTION REPRESENTATION ($F_{StyleCap}$)

In this approach, we generate the artwork representation using the style caption of the artwork. The style information in the style caption describes the style aspects present in an artwork. Formally, we obtain $F_{StyleCap}$ for an artwork $a_i$ as:

$$F_{StyleCap} = T(C(a_i, instruction)) \qquad (1)$$

Where $C$ is the style caption generator and $T$ is the text encoder to encode the caption for the artwork $a_i$. In this work, we leverage an open-source multi-modal large language model(MLLM) called InternVL 2 (Chen et al., 2024). It achieves state-of-the-art performances in most of the validation benchmarks competing with both closed-source proprietary models and other open-source models. It comes in 5 variants and we use the smallest 2 billion parameter model in our work. For the vision

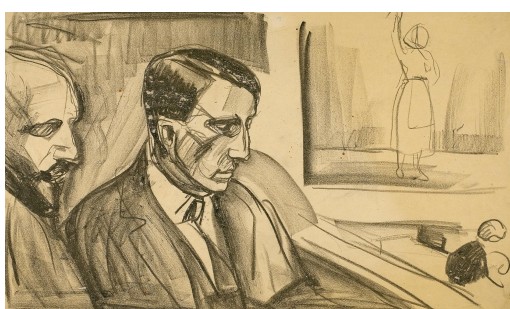 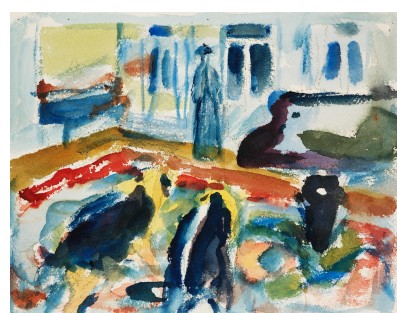

(a) **Style Caption:** *"The style of the given image is reminiscent of early 20th-century European art, possibly influenced by the works of artists like Gustave Moreau or the post-impressionist movement. The use of bold, expressive lines and a monochromatic palette is characteristic of this era. The detailed shading and cross-hatching techniques suggest a focus on texture and depth, creating a sense of three-dimensionality and movement within the composition. The figure in the background, rendered in a simplified manner, adds a sense of context and narrative to the scene. The overall style is evocative of a time when artistic expression was deeply intertwined with the emotional and psychological states of the subject matter."*

(b) **Style Caption:** *"The style of the given image is Abstract Expressionism. This style is characterized by loose, gestural brushwork, vibrant colors, and a focus on the physical act of painting. The painting appears to convey a sense of movement and emotion, with a dynamic interplay of shapes and colors that evoke a lively and energetic atmosphere. The use of bold, contrasting colors and abstract forms suggests a departure from traditional representational art, instead embracing an expressive and personal interpretation of the subject matter.*

Figure 4: Examples of artwork style captions ($F_{StyleCap}$) with the InternVL 2 model for a few artworks from Edvard Munch.

Table 5: Visual elements and style concepts from Kim et al. (2018) utilized for $F_{Annot}$.

| **Visual Elements** | **Concepts** |
| --- | --- |
| Subject | Representational, Non-representational |
| Line | Blurred, Broken, Controlled, Curved, Diagonal, Horizontal, Vertical, Meandering, Thick, Thin, Active, Energetic, Straight |
| Texture | Bumpy, Flat, Smooth, Gestural, Rough |
| Color | Calm, Cool, Chromatic, Monochromatic, Muted, Warm, Transparent |
| Shape | Ambiguous, Geometric, Amorphous, Biomorphic, Closed, Open, Distorted, Heavy, Linear, Organic, Abstract, Decorative, Kinetic, Light |
| Light and Space | Bright, Dark, Medium, Atmospheric, Planar, Perspective |
| General Principles of Art | Overlapping, Balance, Contrast, Harmony, Pattern, Repetition, Rhythm, Unity, Variety, Symmetry, Proportion, Parallel |

part, the 2B variant uses the InternViT model, while for the language part, it uses the Internlm2-chat-1 model (Chen et al., 2024). These models support multiple different modalities like image, text, video, etc., and can handle different outputs such as images, bounding boxes, masks, etc. thereby providing multitask functionality. The InternVL 2 model takes the instruction and the artwork as input and gives us an output text that describes the style of the image. A few examples are showcased in Figure 4.

Next, we use Long-CLIP Zhang et al. (2024) as the text encoder ($T$) for artwork captions. We then use these artwork representations $F_{StyleCap}$ as input artwork representations in the DEC model. It is to be noted that we use InternVL and Long-CLIP as a proxy for $C$ and $T$ respectively and it could be replaced with other image captioning and text encoder models.

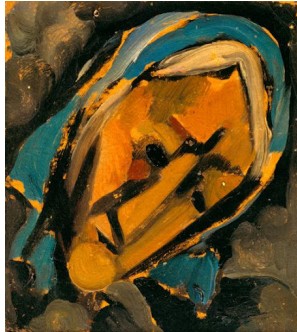

(a) **Style Annotation**
*Subject*: representational
*Line*: blurred, controlled, meandering, thick, thin
*Texture*: smooth, gestural, rough
*Color*: monochromatic
*Shape*: ambiguous, biomorphic, organic, abstract, decorative
*Light and Space*: dark, planar, perspective
*General Principles of Art*: overlapping, balance, contrast, harmony, pattern, repetition

(b) **Style Annotation**
*Subject*: representational, non-representational
*Line*: blurred, controlled, energetic, straight
*Texture*: smooth, gestural, rough
*Color*: cool, warm, muted, chromatic
*Shape*: ambiguous, organic, abstract, decorative
*Light and Space*: bright, dark, atmospheric, planar
*General Principles of Art*: balance, contrast, harmony, pattern, repetition, rhythm, unity, variety, symmetry, proportion, parallel.

Figure 5: Examples of artwork style concept annotation ($F_{Annot}$) with the Intern-VL2 model.

### B.3.2 ARTWORK STYLE CONCEPT ANNOTATION REPRESENTATION ($F_{Annot}$)

In this approach, we annotate artworks with style concepts based on the fundamental principles of art (Ocvirk, 1968). A set of 59 different concepts across seven visual elements has been utilized by Kim et al. (2018) in their work. Formally, we obtain the artwork representation for an artwork $a_i$ with style concept annotation as:

$$F_{Annot} = T(S(a_i, taxonomy, instruction)) \tag{2}$$

where $S$ is the style concept annotator and $T$ is the multi-hot encoder to encode the style concepts of an artwork into a multi-hot vector. The style concept annotator considers the *taxonomy* given in Figure 5 (c) and the *instruction* is to associate the style concepts (for each visual element) from the taxonomy to a given artwork $a_i$. The instruction is constructed in a manner where the instruction includes a query for each style attribute. Similar to $F_{StyleCap}$, we leverage the InternVL 2 as the style concept annotator. After obtaining the style concepts, we turn the 59 style concepts into a multi-hot vector based on whether a style concept is present in the artwork or not. The style information available with this method is fine-grained across various artistic style dimensions. It is to be noted that we use InternVL 2 as a proxy and it could be replaced with other style concept annotators. A few examples of style annotations can be seen in Figure 5.

### B.4 STYLE-TRANSFER BASED STYLE REPRESENTATIONS ($F_{ST}$)

Style transfer is a heavily explored problem where a style-transfer model transfers the style of a style reference image to a content image. Most style-transfer approaches encode the style reference image to obtain style representations which are then used by the model to transfer the style to a content image. In this subsection, we explore different state-of-the-art style-transfer approaches to identify the process used to extract the style representations $F_{ST}$ from a style reference image.

### B.4.1 STYLEGAN BASED STYLE REPRESENTATIONS ($F_{StyleGAN}$)

In StyleGAN (Karras et al., 2019), the generator architecture of the GAN network is modified to include a mapping network and a synthesis network. This architectural design enables disentanglement of the latent factors of variation thereby providing scale-specific control over styles during image synthesis. The mapping network projects the input latent code $z$ of an image to a disentangled intermediate latent space $w$ and the synthesis network, starting the generation from a learned constant vector, leverages these $w$ vectors at different scales to influence the style aspects of the image

by controlling the AdaIN operations after each convolutional layer. In our work, we utilize these $w$ vectors that are responsible for style control and use them as StyleGAN features, $F_{StyleGAN}$.

### B.4.2 STYTR$^2$ BASED STYLE REPRESENTATIONS ($F_{Stytr2}$)

In the Stytr$^2$ (Deng et al., 2022) approach, the style reference image is split into patches. These patches are then passed through a linear projection layer to obtain a sequential representation embedding. The sequential representation embeddings are then passed through a transformer encoder, which consists of a multi-head self-attention block and a feed-forward network. After passing them through the transformer encoder, the output representations extracted from the encoder represent the style information present in an artwork. We term these style representations as $F_{Stytr2}$ representations.

### B.4.3 MAMBA BASED STYLE REPRESENTATIONS ($F_{Mamba}$)

In the Mamba-ST (Botti et al., 2024) approach, similar to Stytr2, both the style reference image and the content image are split into patches and each patch is projected into a 1D embedding using a patch embedding layer. These patch embeddings are then normalized and passed through the domain-specific(content and style) Mamba encoders, followed by an ST-Mamba decoder. When the model is trained using the perceptual and identity losses, the mamba encoders containing base visual state-space machines (VSSMs) learn the domain-specific representations while the ST-Mamba decoder with the help of ST-VSSM learns to fuse the style and content information performing the style-transfer. We leverage the representations from the style mamba encoder since they encompass the style information present in artworks and term them as $F_{Mamba}$.

### B.4.4 STYLESHOT BASED STYLE REPRESENTATIONS ($F_{StyleShot}$)

In the StyleShot (Gao et al., 2024) approach, similar to the previous approaches, the style reference image is split into patches. Unlike the previous approaches, the image is split into multi-scale patches (1/4, 1/8, and 1/16 of an image). For each scale, a distinct ResBlock is utilized to obtain the patch embeddings $f_p$ at each scale. To integrate these multi-level style embeddings, a learnable style embedding $f_s$ is concatenated with the multi-scale embedding ($f_p$) and the combined embedding is fed into a standard transformer. The learnable style embedding ($f_s$) is then extracted from the output of the transformer to obtain a rich style embedding, which we term $F_{StyleShot}$.

### B.4.5 DIFFUSION BASED STYLE REPRESENTATIONS ($F_{DEADiff}$)

We utilize the DEADiff (Qi et al., 2024) model, a diffusion-based text-to-image stylization model that generates images according to the style of a reference image, conditioned on a text prompt. It follows a non-reconstructive learning paradigm to disentangle styles from semantics, by training on paired image datasets that share either the same style or content, but not both, to prevent semantics from getting captured into the style representations. It utilizes Q-Former networks that are instructed to use 'style' and 'content' queries to extract the decoupled representations separately from the reference images. The decoupled representations are then injected into the diffusion's unet at different cross-attention layers through a disentangled conditioning mechanism. The semantic features are injected into coarse, low-level layers, while the style features are injected into fine, high-level layers, ensuring that the style and content are not conflicted in the generated image. We extract the output features of the Q-Former network for style and use them as $F_{DEADiff}$ representations in this work.

### B.5 SPECIAL STYLE-TRAINED IMAGE MODEL BASED REPRESENTATIONS ($F_{Train}$)

Here, we explore the representations extracted from the models that are trained on the datasets with specific style definitions like artwork attribution, Wikiart's artist-based and art movement-based definitions, etc.

### B.5.1 CONTRASTIVE STYLE DESCRIPTORS ($F_{CSD}$)

In this work (Somepalli et al., 2024), the Vision Transformer models (ViT base and large variants) are trained on a multi-label contrastive objective to learn style information from artworks that adhere

to different style attributes. The dataset used for training is curated from LAION Aesthetics by selecting and filtering images according to a predefined set of style tags. The style tags are obtained by combining the bank of artists, mediums, and movement references used on typical user prompts for Stable Diffusion. Each image in the curated dataset can have more than one tag with each tag representing a style attribute. Once the model is trained on this dataset, the representations from the last layer of the ViT backbone are extracted and used as $F_{CSD}$.

### B.5.2 Artwork-trained Image Model based representations

In Chu & Wu (2016; 2018), we observe that training the models on a classification task also makes their representations robust for clustering. Similarly in the case of CSD (Somepalli et al., 2024), we notice that the training dataset used, LAION Aesthetics, contains WikiArt's data as a subset, and the respective representations from the pre-trained model yield higher results in the clustering task than the rest of the representations, as evident from the metric scores. To this end, we venture into this direction by fine-tuning a ViT model on the WikiArt data considering two different ground truth labelings: artist-based and art movement-based.

1. $F_{Artist}$: For the artist-based ground truth dataset, we sorted the artists based on the number of artworks they produced in descending order and selected the artworks from the top 40 artists. We do this to maintain class balance and ensure a sufficient number of samples per each artist class. The total artworks obtained are 25550 which accounts for 32% of the whole WikiArt dataset. Out of this, we use 85% of the data for training and the remaining for testing.

2. $F_{ArtMove}$: Similarly, for the art movement-based ground truth dataset, we sample the same number of artworks (20887 artworks) for the training set as we did for the artist-based data. We use the existing WikiArt subset for the test set.

Using these two different ground truths, we fine-tune two separate models that are pre-trained on Imagenet-21k each for 45 epochs with the cross-entropy loss. We extract the representations from the last layers of the fine-tuned ViT models and use them for clustering.

### B.6 Dimensionality of Representations

In Section B of our paper, we mentioned that we chose global average pooling to reduce the dimensions of our $F_{StyleFeat}$ representations. In this section, we explain in detail our reasoning for choosing the global pooling average as our dimensionality reduction method. The dimensions of the $F_{StyleFeat}$ representations are very high - ranging from 50000 to 200000 approximately. The representation space is too large for efficient clustering; hence, we perform dimensionality reduction before clustering. We utilize Global Average Pooling (GAP) (Lin et al., 2014) as our dimensionality reduction method. We choose GAP over Principal Component Analysis (PCA) (Wold et al., 1987) as it is computationally intensive to apply PCA on a large dataset such as WikiArt (78,978 artworks), and the reduced representations with both methods form similar clusters. We verify this by experimenting with PCA and GAP on the subset of WikiArt dataset and finding that the quantitative results are quite similar to each other as can be seen in Table 6. We use the remaining representations ($F_{ST}$, $F_{Lang}$, and $F_{Train}$) directly for clustering as their dimensionality is sufficiently low. The dimensions before and after using GAP, as well as the respective dimensions of all representations can be seen in Table 6.

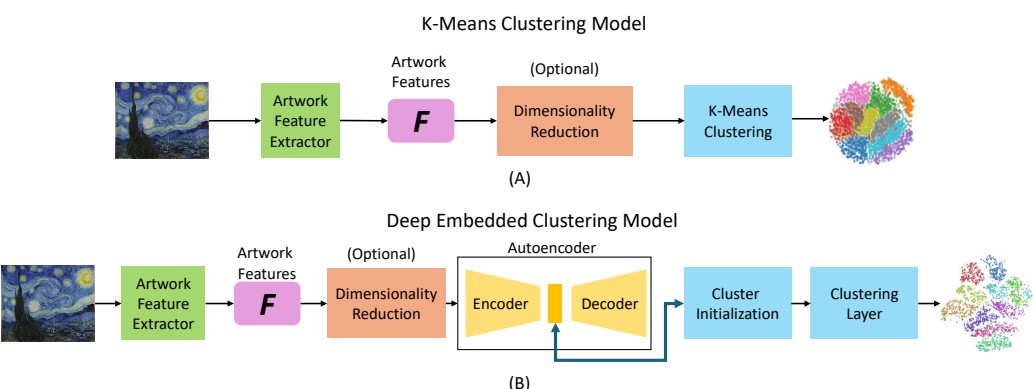

Figure 6: Figures A-B show the architectures which utilize the different representations for style-based clustering which are k-means and deep-embedded clustering models respectively.

Table 6: Summary of the dimensions of different representations and the dimensionality reduction method used is presented on the left. As a few neural representations are high-dimensional, we utilize dimensionality reduction to reduce them to a feasible size for computation. The table on the right showcases the impact of dimensionality reduction methods on the clustering metrics, which was tested on the WikiArt-ArtMove dataset. We observe that using dimensionality reduction techniques minimally affect the clustering results.

| Representation | | Dimensionality Reduction (Yes/No) | Dimensionality Reduction Method | Dimensions before Reduction | Dimensions after Reduction |
|---|---|---|---|---|---|
| $F_{Generic}$ | $F_{Dense}$ | Yes | GAP | 1024x7x7 | 1024 |
| | $F_{LongCLIP}$ | No | N/A | 768 | - |
| | $F_{DINO}$ | Yes | GAP | 257x384 | 384 |
| $F_{StyleFeat}$ | $F_{Gram}$ | Yes | GAP | 512x512 | 512 |
| | $F_{g.c}$ | Yes | GAP | 512x512 | 512 |
| | $F_{IntroStyle}$ | No | N/A | 1280 | - |
| $F_{ST}$ | $F_{StyleGAN}$ | No | N/A | 512 | - |
| | $F_{Stytr2}$ | No | N/A | 512 | - |
| | $F_{Mamba}$ | No | N/A | 512 | - |
| | $F_{Styleshot}$ | No | N/A | 9216 | - |
| | $F_{DEADiff}$ | No | N/A | 12288 | - |
| $F_{Lang}$ | $F_{StyleCap}$ | No | N/A | 768 | - |
| | $F_{Annot}$ | No | N/A | 59 | - |
| $F_{Train}$ | $F_{CSD}$ | No | N/A | 768 | - |
| | $F_{Artist}$ | No | N/A | 1024 | - |
| | $F_{ArtMove}$ | No | N/A | 1024 | - |

| Dimensionality Reduction | SC | CHI |
|---|---|---|
| No Dimensionality Reduction | 0.16 | 595.19 |
| Principal Component Analysis (PCA) | 0.191 | 333.84 |
| Global Average Pooling (GAP) | 0.204 | 405.41 |

## C CLUSTERING MODEL ARCHITECTURES

In this section, we discuss the various clustering architectures we utilize for achieving style-based clustering. For our evaluation framework, we utilize two different clustering architectures:

1. K-Means Clustering architecture
2. Deep Embedding Clustering architecture

Please refer to Figure 6 for an architectural overview of both the clustering architectures.

### C.1 K-MEANS CLUSTERING MODEL

The K-Means clustering model (MacQueen et al., 1967) first initializes a random set of points as cluster centroids which is equal to the number of clusters (K) provided as input. Each data point is assigned to a certain centroid based on which centroid has the smallest Euclidean distance to the specific data point. After all the data points are assigned to a certain centroid, we obtain K clusters where each cluster comprises a number of data points. After obtaining the clusters, a new set of centroids is calculated by averaging all the data points present in each cluster. The process of assigning data points to clusters is repeated again. This process is repeated until there is no change in the assignment of data points to a centroid, which gives us the final set of K clusters.

## C.2 DEEP CLUSTERING MODEL

We briefly discussed the Deep embedded clustering model in Section *Experiments and Evaluation Criteria* of our paper. In this section, we discuss in detail the generalized architecture for Deep Embedded Clustering(DEC) with a deep neural network that produces deep-layer features $F$ for image data, an autoencoder, and a clustering module is presented in Figure 6 (B). The fundamental idea of the DEC method (Xie et al., 2016) is to learn a mapping from the data space to a lower-dimensional feature space which is iteratively optimized with a clustering objective. The model consists of an autoencoder and a clustering layer connected to the embedding layer of the autoencoder.

**Autoencoder:** Autoencoders are deep neural networks that can project the input data into latent space using an encoder and reconstruct the original input from latent space using a decoder. The encoder present in the autoencoder first takes the input data and transforms the data with a non-linear mapping $\phi_\theta : X \to Z$ where X is the input space of the data and Z is the hidden latent space. The decoder learns to reconstruct the original input based on the latent representation, $\psi : Z \to X$. The latent embedded features are then propagated through the decoder so it can reconstruct the latent features back to the original input space. The non-linear mapping of $\phi$ and $\psi$ is learnt by updating the autoencoder parameters by minimizing a classic mean squared reconstruction loss:

$$L_r = \frac{1}{n} \sum_{i=1}^{n} ||x_i' - x_i||^2 = \frac{1}{n} \sum_{i=1}^{n} ||\psi(\phi(x_i)) - x_i||^2 \tag{3}$$

where n is the cardinality of the input features, $x_i$ is the *i*-th input sample, $x_i'$ is the reconstruction performed by the decoder and $|| \cdot ||$ is the Eucledian Distance.

**Clustering Layer:** The clustering layer takes the latent embedded features from the encoder based on the non-linear mapping $\phi : X \to Z$ and initially assigns each embedded point to $k$ cluster centroids by using k-means clustering $\{c_j \in Z\}_{j=1}^{k}$ where $c_j$ represents the *j*th cluster centroid.

After the initialization, each embedded point, $z_i = \phi(x_i)$ is mapped to a cluster centroid $c_j$ by using a cluster assignment $Q$ based on Student's t-distribution:

$$q_{ij} = \frac{(1 + ||z_i - c_j||^2)^{-1}}{\sum_{j'}(1 + ||z_i - c_{j'}||^2)^{-1}} \tag{4}$$

where $j'$ represent every cluster and $q_{ij}$ represents the membership probability of $z_i$ to belong to the cluster *j* which basically soft assigns $z_i$ to cluster centroid $c_j$. $q_{ij}$ represents the similarity between a datapoint $z_i$ and the cluster centroid $c_j$ which gives us the confidence of a datapoint being assigned to a particular cluster.

The decoder of the autoencoder is abandoned and the DEC model jointly optimizes clustering layer and encoder based on the auxiliary target distribution $p_{ij}$ calculated from $q_{ij}$ derived from Eq.4 which emphasizes the data points that have higher confidence assigned to them while also minimizing the loss contribution of each centroid:

$$p_{ij} = \frac{q_{ij}^2/f_j}{\sum_{j'} q_{ij'}^2/f_{j'}} \tag{5}$$

where $f_j = \sum_j q_{ij}$ are the soft cluster frequencies. The DEC model optimizes the target function by minimizing the Kullback-Leibler(KL) divergence between P and Q where P is the auxiliary target function defined in Eq.5 and Q is the cluster assignment based on Student's t-distribution. This improves the initial cluster estimate by learning from previous high-confidence predictions.

$$L_c = KL(P||Q) = \sum_i \sum_j p_{ij} \log\left(\frac{p_{ij}}{q_{ij}}\right) \tag{6}$$

The cluster centers $c_j$ and the encoder parameters $\theta$ (of autoencoder) are then jointly optimized using Stochastic Gradient Descent (SGD) with momentum (Xie et al., 2016).

**Implementation Details:** We conduct our experiments on A100 GPU with 15 GB RAM and 5GB VRAM. The deep embedded clustering model is trained using the Adam optimizer. The number of iterations are set to 8000 and the convergence threshold is set to 0.0001.

We test with varying the DEC encoder's final layer's size (refer to Table 7) to see if it has an effect on the features. We observe that the there is minimal change in the cluster ability of the features when the they are encoded to different sized latents.

Table 7: Quantitative results for the WikiArt-ArtMove dataset when the size of final layer of the encoder used in DEC is varied. We observe minimal change in the results even when we change the final layer of the encoder.

| Encoder Final Layer Size | SC | CHI |
|---|---|---|
| 10 | 0.118 | 8059.72 |
| 50 | 0.102 | 226.7 |
| 100 | 0.204 | 465.31 |

### C.3 Impact of DEC Clustering on Style-Based Clustering

The choice of clustering architecture plays a crucial role in style-based clustering performance. DEC clustering directly influences the latent representations assigned to each cluster by adaptively modifying input representations when the original features fail to form adequate clusters. This modification process enables DEC to better differentiate between representation-specific features, ultimately producing more distinct and well-separated clusters.

Our analysis of Figure 7 reveals a consistent pattern across DEC iterations: internal evaluation metrics (Silhouette Coefficient and Calinski-Harabasz Index) show continuous improvement, while external metrics (ARI: Adjusted Rand Index, NMI: Normalized Mutual Information) remain relatively stable. This phenomenon occurs consistently across different neural style representations, including $F_{StyleCap}$, $F_{Gram}$, and $F_{StyleShot}$.

The divergence between internal and external metric performance indicates that DEC successfully enhances the geometric quality of clusters—creating more cohesive and well-separated groups—without necessarily improving the semantic alignment with ground-truth style categories. This suggests that while DEC optimizes cluster structure in the representation space, the enhanced clustering may not directly translate to better style-based groupings.

## D Additional Details on Evaluation Metrics

**Internal Evaluation Metrics**

- **Silhouette Coefficient (SC)** (Rousseeuw, 1987): SC is the measure of how similar a data point is to other data points in its own cluster and how similar the same data point is to the data points in a separate cluster. This metric is calculated on the data point level. SC ranges from -1 to +1, where a high value indicates that the data points are well-matched to their own clusters and poorly matched to other clusters. A lower value would indicate that the data points are wrongly assigned to clusters. The SC for a single sample is given by:

$$SC = \frac{b - a}{max(a, b)} \tag{7}$$

  where $a$ is the distance between the sample and the closest data point in the *same cluster*, whereas $b$ is the distance between the sample and the closest data point in a *different cluster*. The SC is calculated as the average of the SC associated with every data point.

- **Calinski-Harabasz Index (CHI)** (Caliński & JA, 1974): CHI is the ratio of the sum of intra-cluster dispersion and inter-cluster dispersion for all clusters. This metric is calculated at the cluster level. A higher value of *CHI* indicates that the clusters are more spread out and dense. The CHI is given by:

$$CHI = \frac{BCSS}{WCSS} \times \frac{n - k}{k - 1} \tag{8}$$

  where $k$ is the number of clusters, $n$ is the total number of samples, and $BCSS$ (Between Clusters Sum of Squares) is the weighted sum of squared Euclidean distances between all

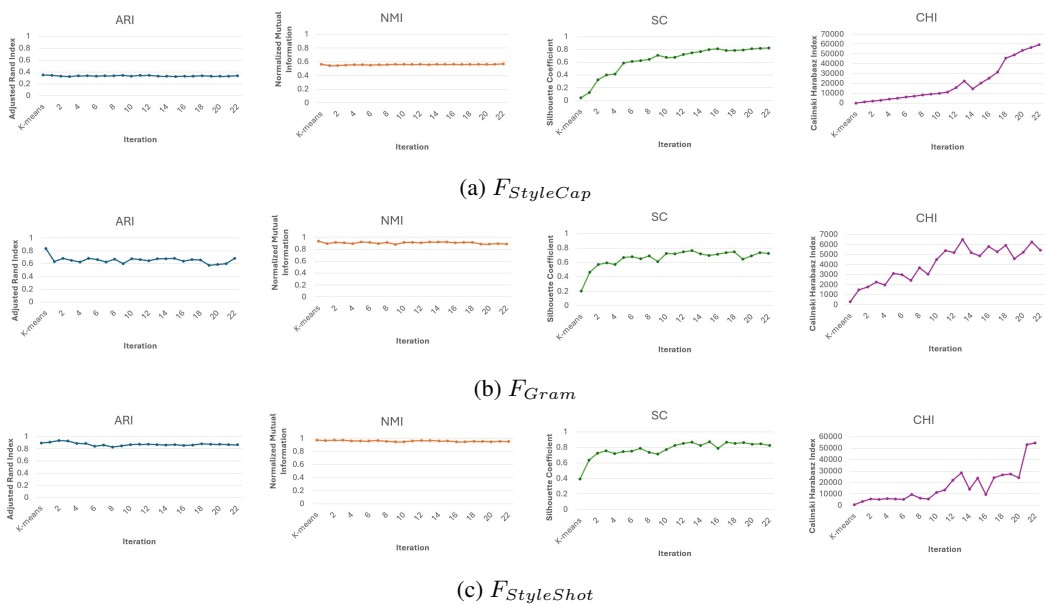

(a) $F_{StyleCap}$

(b) $F_{Gram}$

(c) $F_{StyleShot}$

Figure 7: Evolution of evaluation metrics across DEC iterations for different neural style representations on the Mixed StyleShot Curated dataset (MSC). The initial iteration represents K-means initialization, with subsequent iterations corresponding to DEC model updates (labels updated every 140 training iterations). Internal metrics (SC:Silhouette Coefficient and CHI:Calinski-Harabasz Index) consistently improve across iterations, indicating enhanced cluster cohesion and separation in the representation space. However, external metrics (ARI: Adjusted Rand Index, NMI: Normalized Mutual Information) remain stable, suggesting that while DEC optimizes geometric cluster quality, the representation modifications do not improve semantic alignment with ground-truth style categories.

the cluster centroids and the dataset centroid. It is given by $BCSS = \sum_{i=1}^{k} n_i ||c_i - c||^2$, where for a cluster $i$, $n_i$ is the number of data points in a cluster, $c_i$ is the centroid of the cluster, and $c$ is the centroid of the entire data. $WCSS$ (Within Clusters Sum of Squares) is the sum of squared Euclidean distance between all the data points in a cluster and their respective cluster centroid. It is given by $WCSS = \sum_{i=1}^{k} \sum_{x \in C_i} ||x - c_i||^2$ where for a cluster $i$, $C_i$ represents a cluster and $x$ is a data point that belongs to cluster $C_i$.

**External Evaluation Metrics**

- **Adjusted Rand Index (ARI)** (Rand, 1971): Rand Index (RI) is a measure of similarity between two data clusterings. It takes all pairs of samples from both ground truth and predicted clusterings and considers all pairs of agreements and disagreements in their assignments to clusters. It then adjusts the index to account for change by taking into account the expected similarity between the two clusterings. The Rand Index score ranges from -1 to 1. Values ranging between -1 to 0 indicate disagreement between the two data clusterings whereas values ranging from 0 to 1 indicate agreement between the two data clusterings. The Rand Index (RI) is given by

$$RI = \frac{(x + y)}{\frac{n}{2}} \tag{9}$$

where $x$ represents the number of data points belonging to the same cluster in both the clusters obtained from the clustering model and the ground truth, whereas $y$ denotes the number of data points that belong to different clusters across both the ground truth and the clusters produced by the clustering model. $n$ denotes the total number of data points. Since RI does not take the random assignment of data points into consideration, we use

ARI which accounts for this chance of random assignments. ARI is given by:

$$ARI = \frac{RI - E}{Max(RI) - E} \tag{10}$$

where $RI$ is the Rand Index, $E$ is the expected Rand Index and $Max(RI)$ is the maximum value that $RI$ can take (always 1).

- **Normalized Mutual Information (NMI)** (Shannon, 1948): Mutual Information (MI) is used to calculate the information shared between the ground truth clustering and the predicted clustering. The MI ranges from 0 to 1, where a value closer to 0 would indicate no correlation between ground truth and predicted clusters whereas a value closer to 1 would indicate a near-perfect correlation between ground truth and predicted clusters. The MI is given by:

$$MI = \sum_{i=1}^{|U|} \sum_{j=1}^{|V|} \frac{|U \cap V|}{N} \log \frac{N|U \cap V|}{|U||V|} \tag{11}$$

where $U$ and $V$ are the predicted clustering and the ground truth clustering respectively, $|U|$ and $|V|$ are the number of samples in both the clusterings and $N$ is the total number of samples. Since MI is not normalized, the calculated value of MI might be higher due to the imbalance of cluster distribution in the datasets. Hence we use Normalized Mutual Information, which normalizes the Mutual Information and is given by:

$$NMI = \frac{2 \times MI}{[H(U) + H(V)]} \tag{12}$$

where $MI$ is the mutual information and $H$ is the entropy of the respective clustering.

## E  ADDITIONAL DETAILS ON DATASETS

In this section, we present a few additional details on the datasets we use for our experiments. In our experiments, we use four different datasets with different style definitions:

1. WikiArt Art Movement based dataset (WikiArt-ArtMove): For this dataset, we utilize the complete WikiArt collection (WikiArt.org, 2010; WikiArt) of 78,978 artworks categorized into 27 art movements. A few samples can be seen in Figure 8 (a).

2. WikiArt Artist-based dataset (WikiArt-Artist): We pick 25,550 artworks from the top 40 artists with the highest number of artworks from the WikiArt art dataset and categorize them based on the artist who created a certain artwork. A few samples can be seen in Figure 8 (a).

3. DomainNet dataset: We create a subset of the DomainNet dataset (Peng et al., 2019) by taking 10 images from each style class (6 classes) for 50 content classes. In total, we obtain 3000 images from 50 content classes and 6 style classes. A few samples can be seen in Figure 8 (b).

4. Synthetically Curated datasets: By leveraging style-transfer techniques, we create two synthetically curated datasets called Mixed StyleShot Curated dataset (MSC) and Mixed Mamba Curated dataset (MMC), created by using StyleShot (Gao et al., 2024) and Mamba-ST (Botti et al., 2024) respectively. We utilize 50 content images from the MSCOCO Lin et al. (2015) dataset and pick 10 style images from four datasets: the WikiArt-ArtMove dataset, the Edvard Munch Archive, the Brueghel dataset, and the Clip Art Illustrations dataset.

   - **Edvard Munch Archive (EMA)**: We experiment with the artwork collection dedicated to the artist Edvard Munch. We specifically consider sketch and watercolor paintings, comprising 7410 artworks created by Edvard Munch (Sivertsen et al., 2023). The artworks are categorized based on shading and color. We present a few examples in 9 (a).

   - **Brueghel Dataset (BD)**: The Brueghel dataset (Shen et al., 2019) consists of 1587 artworks created by Jan Brueghel the Elder. This dataset consists of artworks in different media like oil, ink, and watercolor, along with various painting surface materials such as paper, panel, and copper. We present a few examples in 9 (b).

- **Clip Art Illustrations Dataset (CAID)**: Clip art images consist of various styles such as sketches, woodcuts, cartoons, and gradient-shading. We adopt the clip art illustrations dataset used in Garces et al. (2014), consisting of 4591 clip art illustrations. 1000 of the illustrations have been collected from the Art Explosions dataset (Art-Explosion, 2024), and 3591 of those illustrations are from the clip art included in Microsoft Office. We present a few examples in 9 (c).

We present a few samples from curated datasets created using StyleShot and Mamba-ST in Figure 8 (c) and (d).

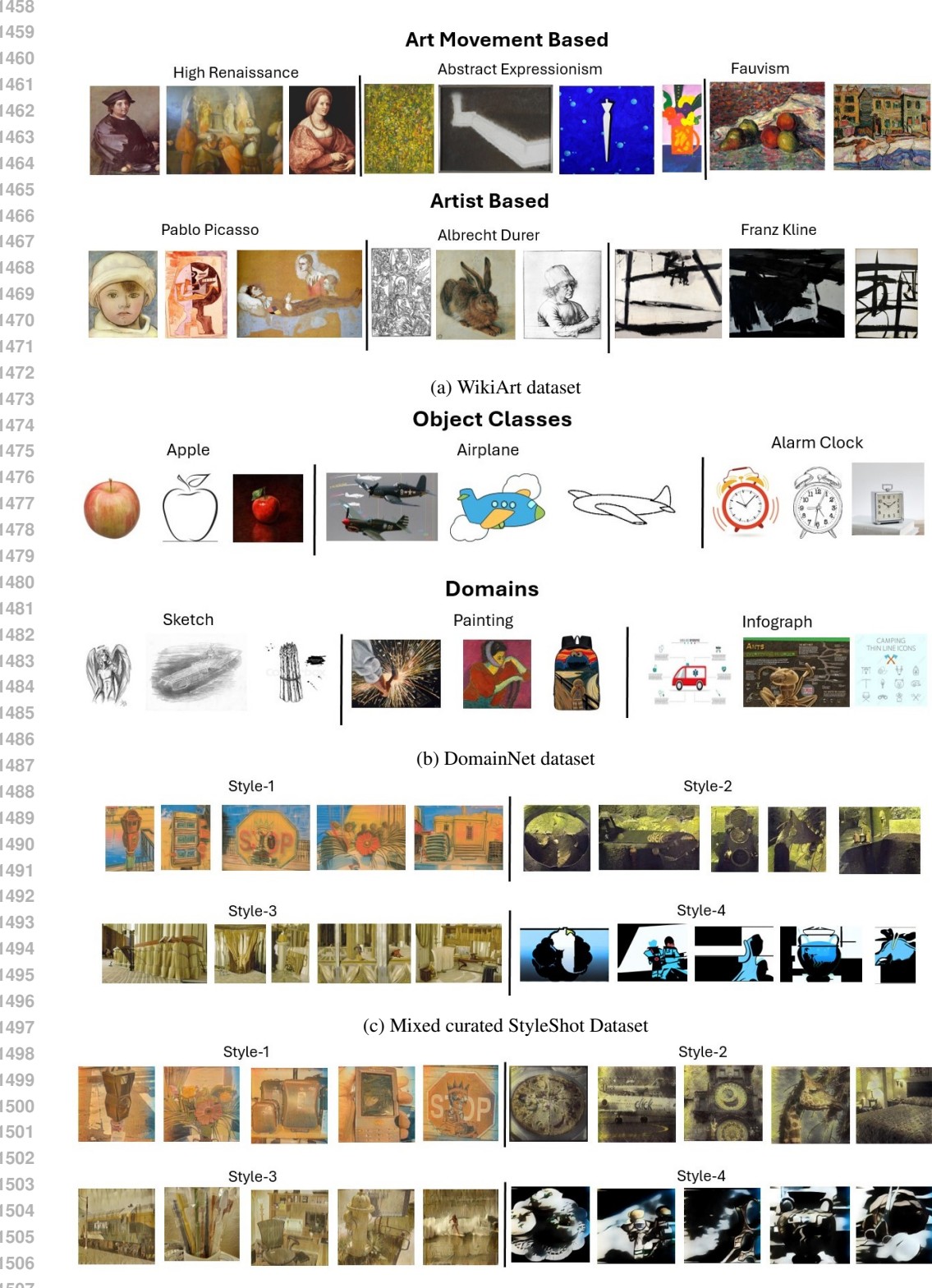

(a) WikiArt dataset

(b) DomainNet dataset

(c) Mixed curated StyleShot Dataset

(d) Mixed curated Mamba-ST dataset

Figure 8: Representative samples for the (a) WikiArt-ArtMove and WikiArt-Artist, (b) DomainNet, (c) Mixed curated StyleShot Dataset and (d) Mixed curated Mamba-ST dataset. (Please zoom in for finer details)

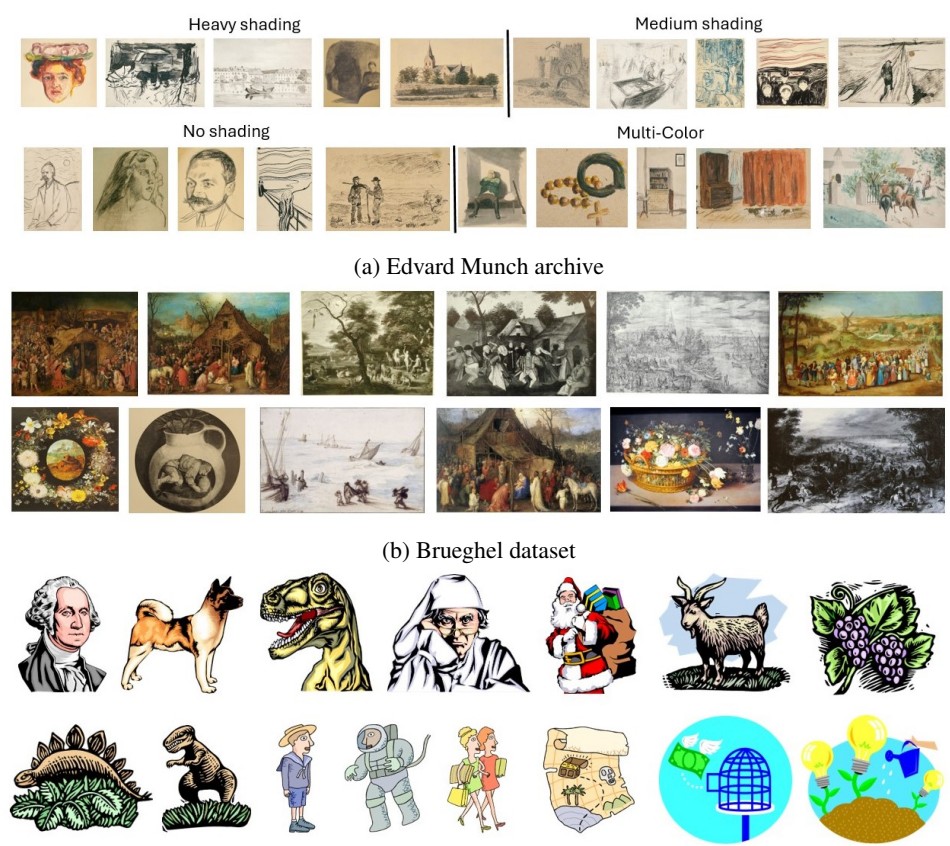

Figure 9: Representative samples from the additional datasets (a) Edvard Munch archive, (b) Brueghel dataset and (c) Clip-art illustrations dataset. (Please zoom in for finer details).

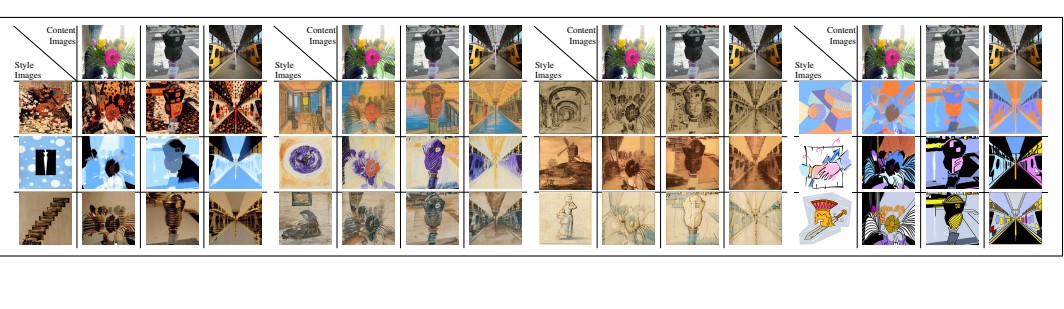

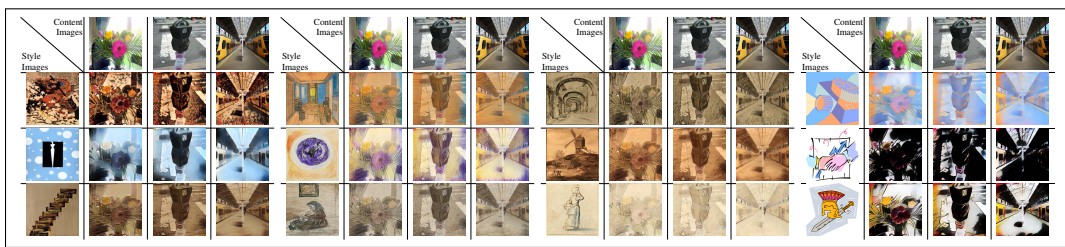

Figure 10: Examples of content images and style images and their respective style-transfer output images from Styleshot (top row) and Mamba-ST (bottom row). The content images were picked from the MS-Coco dataset (Lin et al., 2015) and the style images were picked from WikiArt dataset (column 1), Munch dataset (column 2), Brueghel dataset (column 3) and Clip-art illustrations dataset (column 4). (Please zoom in for finer details).

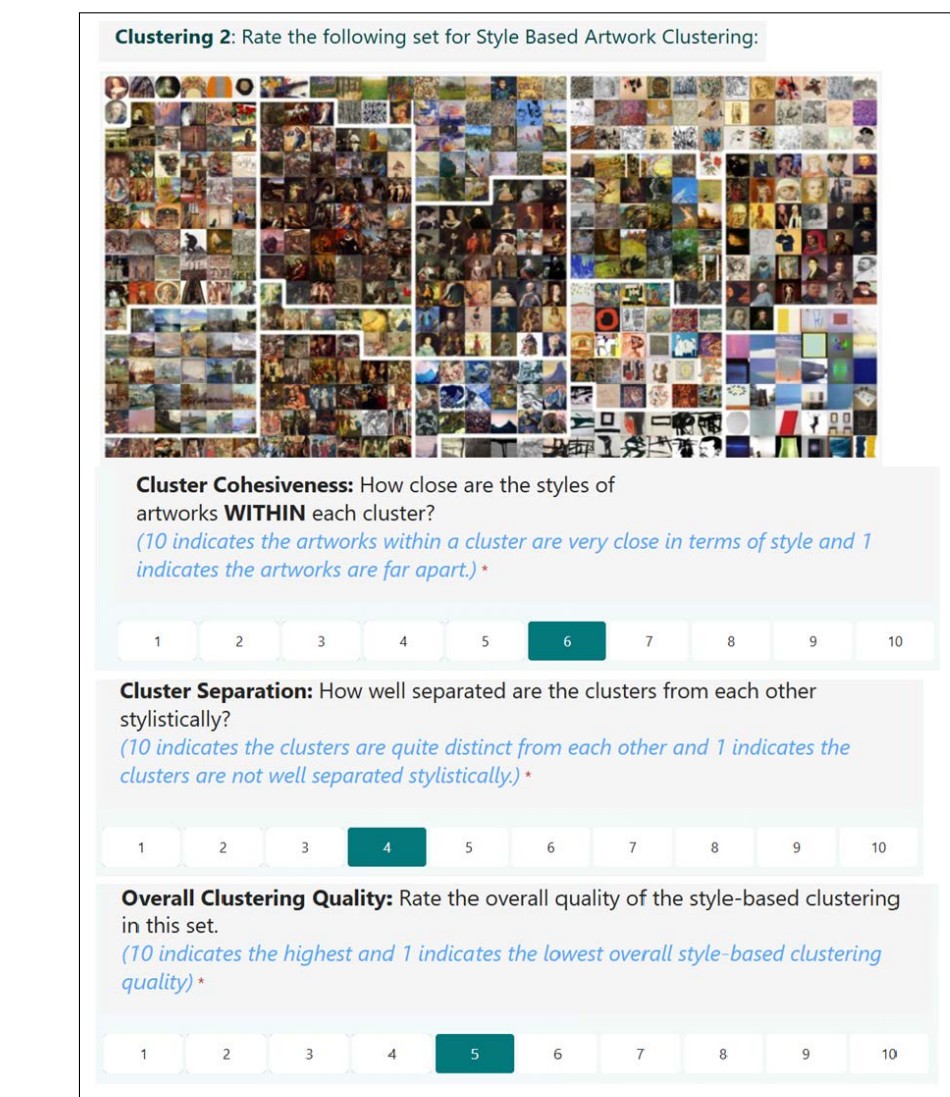

Figure 11: Screenshot of the questions asked for a specific clustering in the human perception survey.

## F    HUMAN PERCEPTION SURVEY DETAILS

Comprehensive human evaluation of clustering quality across multiple neural representations presents significant scalability challenges. While we primarily rely on quantitative metrics, we conducted a targeted human perception study to assess clustering quality from a cognitive perspective. Our study involved 25 participants with visual arts backgrounds, as detailed in the main paper.

Each participant evaluated 5 high-resolution clustering visualizations representing different clustering approaches, presented in randomized order without method identification. Participants rated each clustering result on a 1-10 scale across three dimensions (see Figure 11 for example stimuli):

- **Q1. Intra-cluster Cohesiveness:** How stylistically similar are artworks within individual clusters?
- **Q2. Inter-cluster Separation:** How stylistically distinct are the clusters from one another?
- **Q3. Overall Quality:** What is the overall effectiveness of the style-based clustering?

Participants were instructed that clustering was performed based on artistic style, with other visual elements (e.g., content, subject matter) considered relevant only insofar as they contribute to stylistic

characteristics. Importantly, we provided no explicit definition of "style," allowing participants to apply their own aesthetic understanding and perceptual frameworks.

The survey results revealed distinct preferences among clustering methods. Participants consistently rated $F_{StyleShot}$ from the $F_{ST}$ category as producing the highest quality and most cohesive clusters, despite somewhat lower inter-cluster separation scores. Across all methods, inter-cluster separation received the most variable and generally lowest ratings, suggesting this aspect of clustering is most challenging to achieve perceptually.

Notably, even while the ground truth WikiArt movement-based clustering was included as one of the five methods, participants did not rate it as the optimal style clustering. This finding highlights potential discrepancies between art historical categorizations and contemporary perceptual judgments of stylistic similarity, suggesting that human-perceived style relationships may diverge from established art movement taxonomies.

These results point toward the importance of perception-driven approaches to understanding artistic style, representing a promising direction for future investigation in computational aesthetics and style analysis.

Table 8: Quantitative results and indicative qualitative rating for **content-based clustering (left)** and **style-based clustering (right)** on the *DomainNet* dataset. The dataset includes 3000 images from 6 style classes and 50 content classes. For both content and style-based clustering $F_{Dense}$, the generic representation performs poorly, whereas the style-specific representations perform adequately for content but perform really well for style.

| Representations | ARI | NMI | Qualitative Rating | Representations | ARI | NMI | Qualitative Rating |
|---|---|---|---|---|---|---|---|
| $F_{Dense}$ | 0.106 | 0.364 | Poor | $F_{Dense}$ | 0.291 | 0.352 | Poor |
| $F_{StyleCap}$ | 0.15 | 0.435 | Poor | $F_{StyleCap}$ | 0.547 | 0.591 | Good |
| $F_{Mamba}$ | 0.012 | 0.179 | Very Poor | $F_{Mamba}$ | 0.514 | 0.561 | Fair |
| $F_{DEADiff}$ | 0.06 | 0.33 | Very Poor | $F_{DEADiff}$ | 0.733 | 0.736 | Very Good |
| $F_{CSD}$ | 0.116 | 0.401 | Poor | $F_{CSD}$ | 0.654 | 0.681 | Good |
| $F_{LongCLIP}$ | 0.298 | 0.616 | Poor | $F_{LongCLIP}$ | 0.427 | 0.529 | Fair |
| $F_{DINO}$ | 0.108 | 0.405 | Poor | $F_{DINO}$ | 0.517 | 0.568 | Fair |

# G  QUALITATIVE COMPARISON AND ADDITIONAL QUANTITATIVE RESULTS

In this section, we present the additional details for the results in our main paper which could not be included in our main paper.

## G.1  DOMAINNET DATASET: QUANTITATIVE AND QUALITATIVE RESULTS

In Table 8, we showcase the quantitative results for the DomainNet dataset for both content and style clustering utilizing the style representation from each category along with a generic representation ($F_{Dense}$). Both content and style based clusterings were obtained by setting the number of clusters to be equal to either the number of content classes or the style classes. As mentioned in our main paper, we observe that the generic representations perform adequately for both content as well as style clustering. For style-clustering, we observe that the style-specific representations out perform $F_{Dense}$ by a large margin. This is further reinforced when we look at Figure 13 which showcase the color representations across all feature representations for content and style clustering. We further showcase qualitative result for content and style clustering with $F_{StyleCap}$ in Figure 12 which also showcase the same behavior.

## G.2  WIKIART-ARTMOVE DATASET: QUALITATIVE EVALUATION

For the art movement WikiArt dataset, we present the ground truth in Figure 14. In Figures 15, 16, 17 and 18 we present the qualitative results along with their color representation for the WikiArt-ArtMove dataset with the representative feature representations from each category of style-based representations. We observe that for the art movement style definition, all 4 representations perform poorly in terms of ground truth, but when observed visually, the cluster formed through the style-based representations are similar in terms of style. This is further support by the human survey results in 2, where the participants preferred the $F_{StyleShot}$ representation clusters over the WikiArt-ArtMove ground truth clusters.

## G.3  WIKIART-ARTIST DATASET: QUALITATIVE EVALUATION

In Figure 19, we present the ground truth associated with the artist-based style definition for the WikiArt-Artist dataset. We then present the qualitative results of different representations on this dataset in Figures 20, 21, 22 and 23. We observe that the most of the style representations apart from $F_{CSD}$ perform poorly on style-clustering based on artists. As $F_{CSD}$ is trained on multiple artwork labels like the artist, it's able to perform well in creating distinct style-based clusters.

## G.4  SYNTHETICALLY CURATED DATASETS: QUANTITATIVE AND QUALITATIVE EVALUATION

In our main paper, we presented the quantitative results with the synthetically curated dataset created using StyleShot. In Table 9, we showcase the quantitative results on the curated dataset obtained through Mamba-ST (MMC). We observe a similar trend as the MSC dataset of style-transfer based

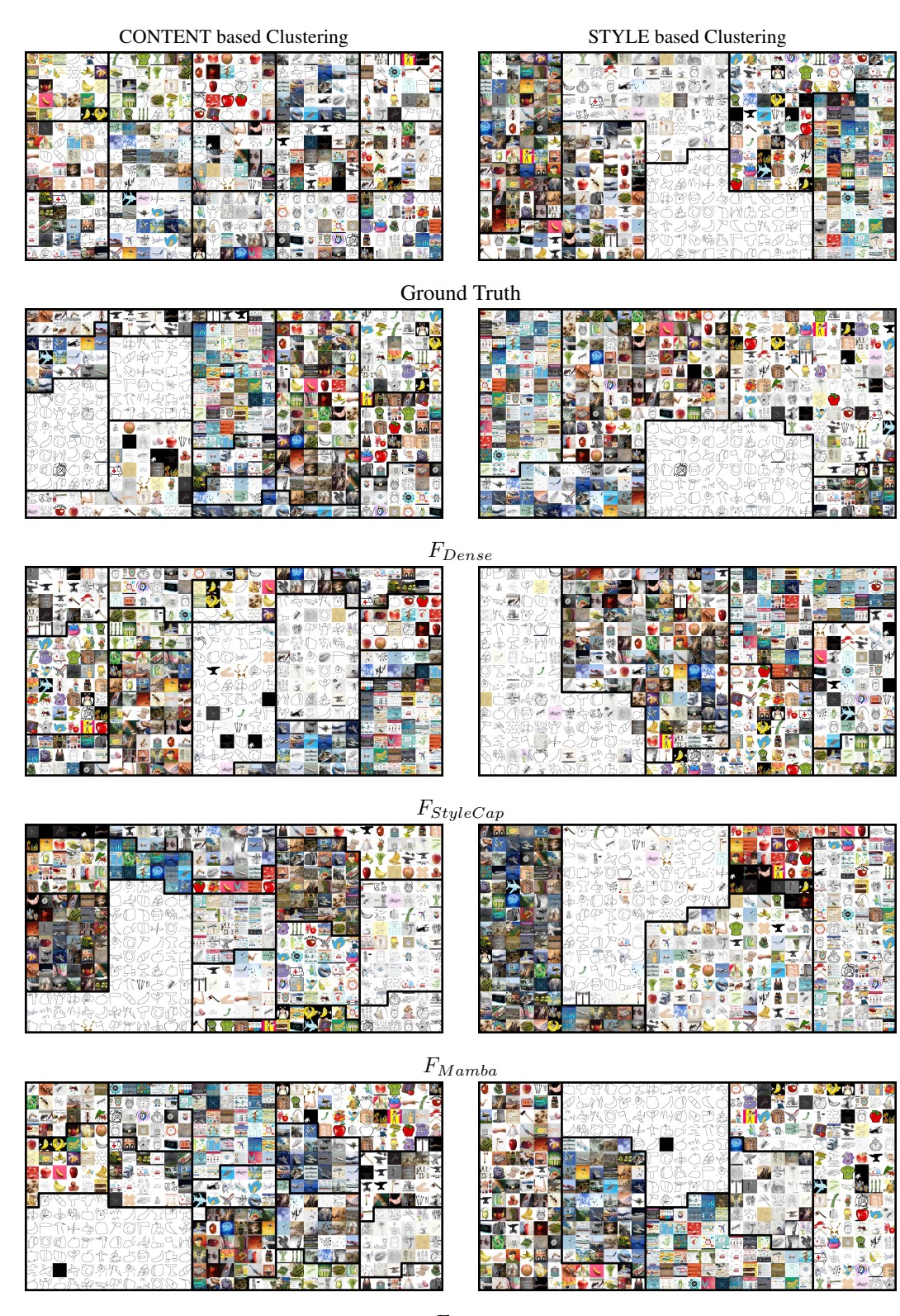

Figure 12: Qualitative comparison of style-based and content-based clustering through the select four neural feature representations on the *DomainNet* dataset.

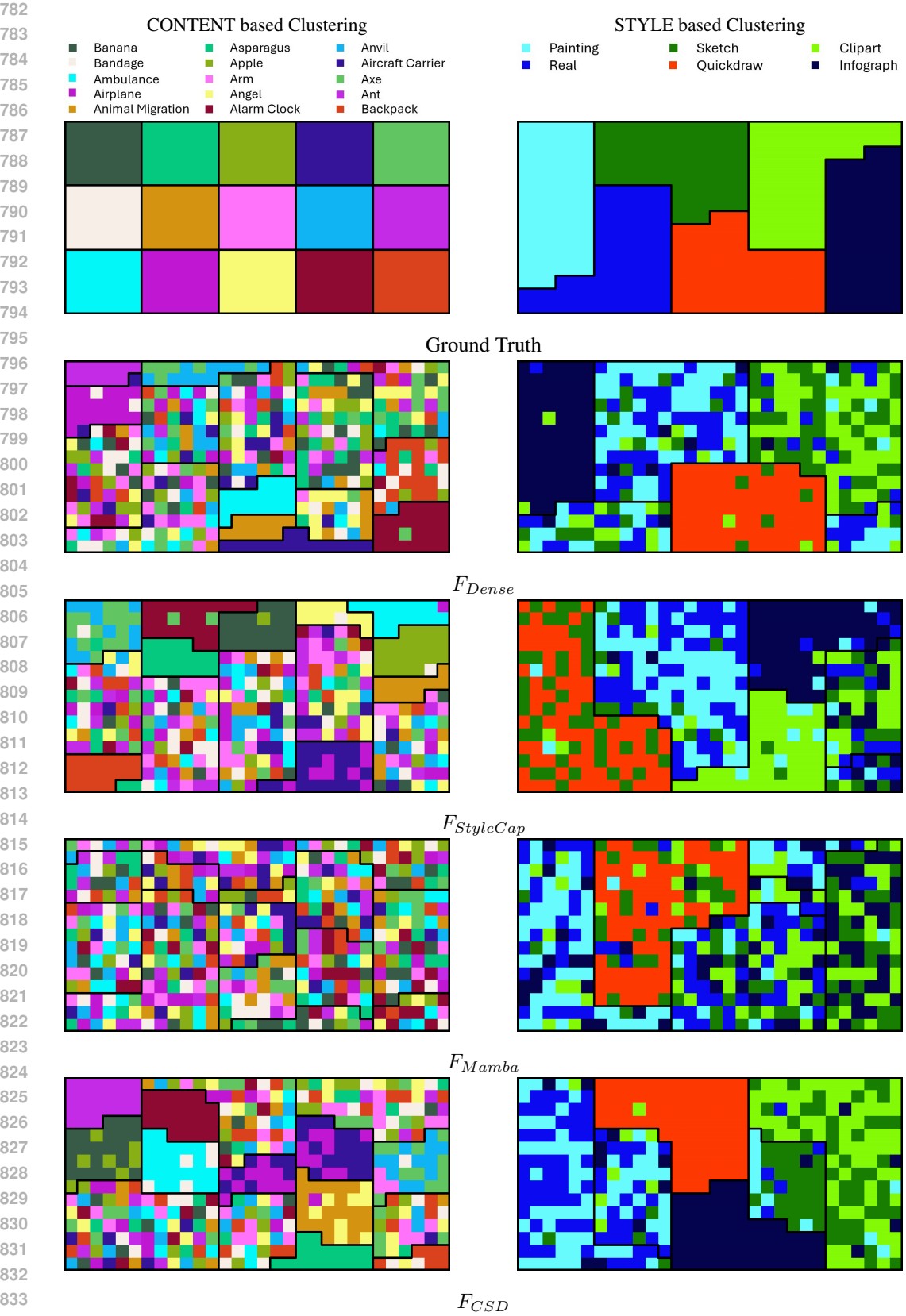

Figure 13: Visual comparison of the relative effectiveness of style-based and content-based clustering through the select four neural feature representations on the *DomainNet* dataset. For perfect clustering, each cluster would have a distinct and homogeneous color patches.

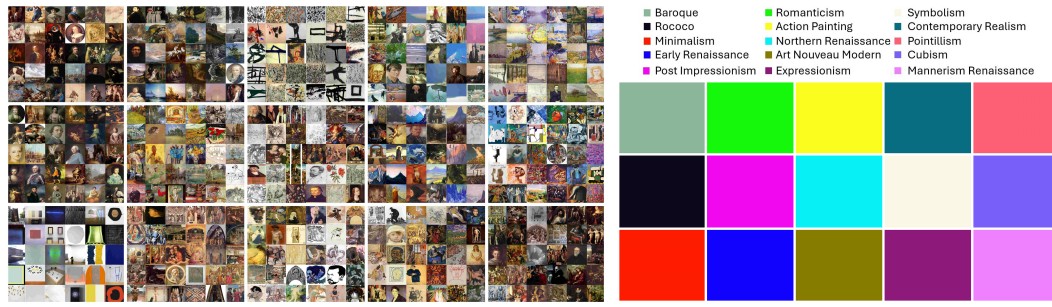

(a) *Ground Truth* style clusters      (b) Class distribution in *Ground Truth* clusters

Figure 14: Artwork samples from the *WikiArt-ArtMove* dataset serving as the ground truth for qualitative comparison of clustering with different neural representations.

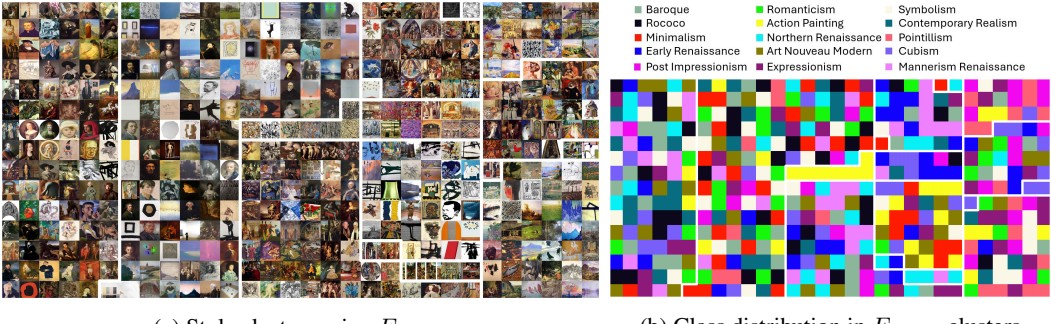

(a) Style clusters using $F_{Gram}$      (b) Class distribution in $F_{Gram}$ clusters

Figure 15: Qualitative results of style clustering on the sample *WikiArt-ArtMove* dataset (Fig. 14) using $F_{Gram}$ (category: $F_{Class}$) neural style representation.

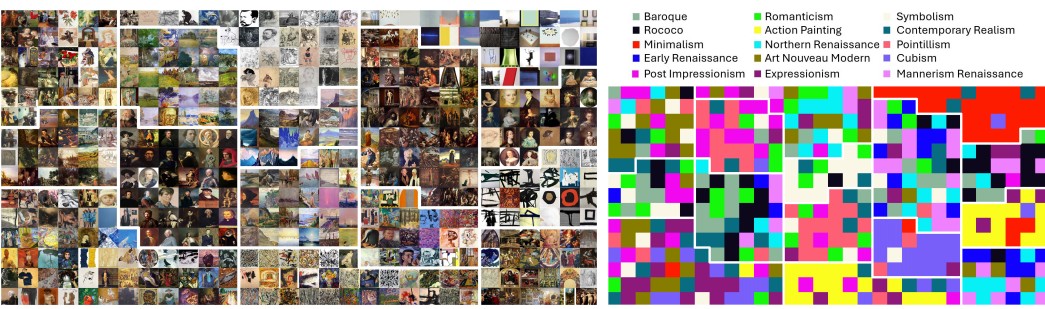

(a) Style clusters using $F_{StyleCap}$      (b) Class distribution in $F_{StyleCap}$ clusters

Figure 16: Qualitative results of style clustering on the sample *WikiArt-ArtMove* dataset (Fig. 14) using $F_{StyleCap}$ (category: $F_{Lang}$) neural style representation.

representations performing the best for this style definition. We also present the qualitative results for the MSC in Figures 25, 26, 27 and 28, where both $F_{Stytr2}$ and $F_{CSD}$ show almost perfect clustering based on ground truth.

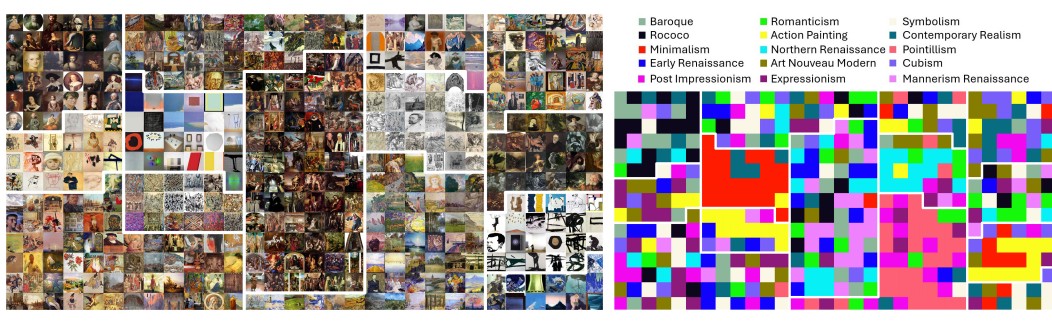

(a) Style clusters using $F_{StyleShot}$      (b) Class distribution in $F_{StyleShot}$ clusters

Figure 17: Qualitative results of style clustering on the sample *WikiArt-ArtMove* dataset (Fig. 14) using $F_{StyleShot}$ (category: $F_{ST}$) neural style representation.

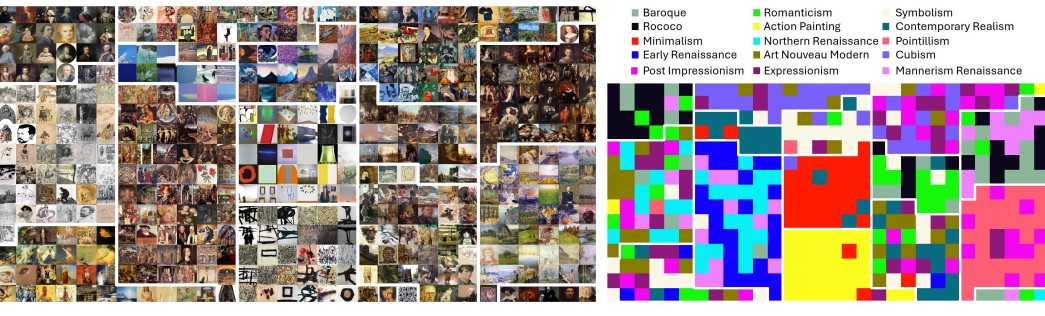

(a) Style clusters using $F_{CSD}$      (b) Class distribution in $F_{CSD}$ clusters

Figure 18: Qualitative results of style clustering on the sample *WikiArt-ArtMove* dataset (Fig. 14) using $F_{CSD}$ (category: $F_{Train}$) neural style representation.

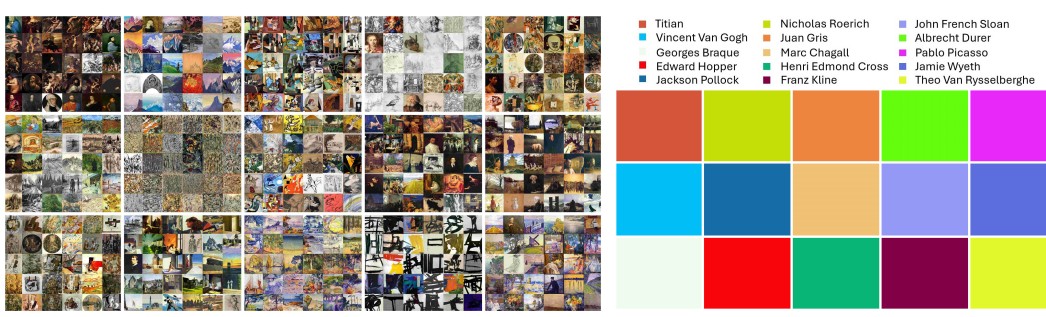

(a) *Ground Truth* style clusters

(b) Class distribution in *Ground Truth* clusters

Figure 19: Artwork samples from the *WikiArt-Artist* dataset serving as the ground truth for qualitative comparison of clustering with different neural representations.

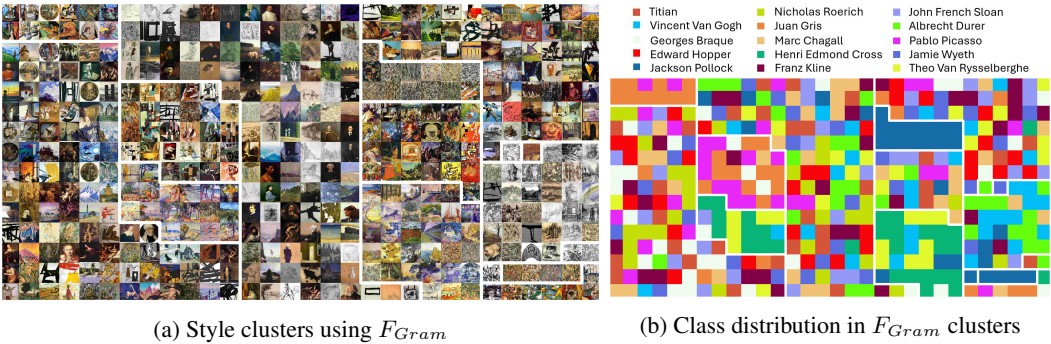

(a) Style clusters using $F_{Gram}$

(b) Class distribution in $F_{Gram}$ clusters

Figure 20: Qualitative results of style clustering on the sample *WikiArt-Artist dataset* (Fig. 19) using $F_{Gram}$ (category: $F_{StyleFeat}$) neural style representation.

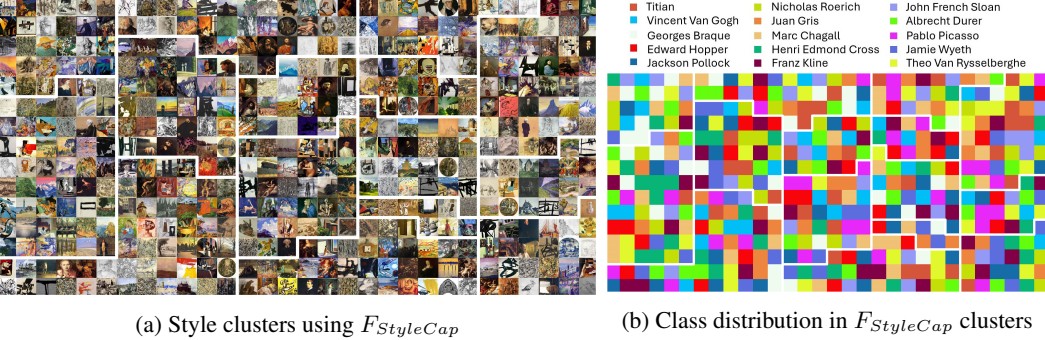

(a) Style clusters using $F_{StyleCap}$

(b) Class distribution in $F_{StyleCap}$ clusters

Figure 21: Qualitative results of style clustering on the sample *WikiArt-Artist dataset* (Fig. 19) using $F_{StyleCap}$ (category: $F_{Lang}$) neural style representation.

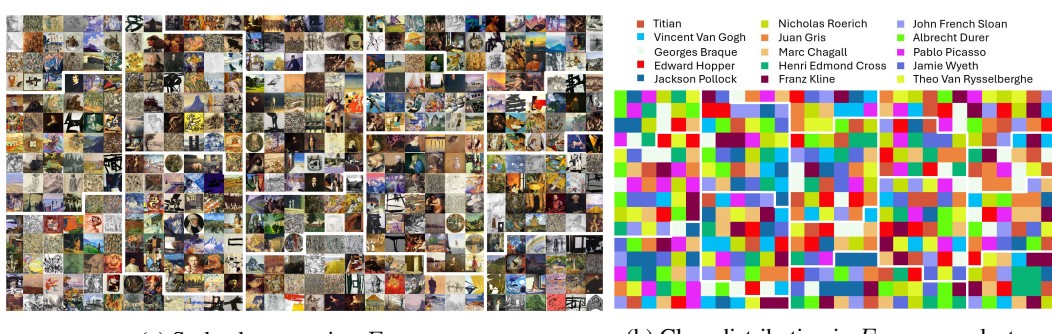

(a) Style clusters using $F_{StyleShot}$       (b) Class distribution in $F_{StyleShot}$ clusters

Figure 22: Qualitative results of style clustering on the sample *WikiArt-Artist dataset* (Fig. 19) using $F_{StyleShot}$ (category: $F_{ST}$) neural style representation.

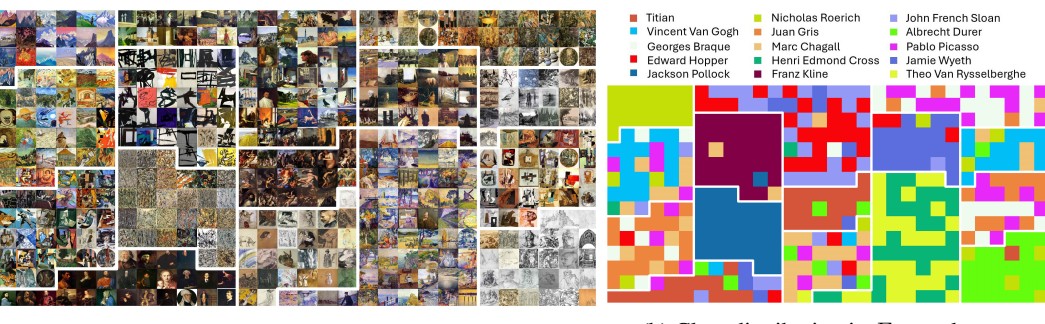

(a) Style clusters using $F_{CSD}$       (b) Class distribution in $F_{CSD}$ clusters

Figure 23: Qualitative results of style clustering on the sample *WikiArt-Artist dataset* (Fig. 19) using $F_{CSD}$ (category: $F_{Train}$) neural style representation.

Table 9: Metrics scores for the **Mixed curated** dataset created using **Mamba-ST** (MMC) for all features for both K-Means and DEC model. The best , second best , third best and the worst results are highlighted for each metric. MMC contains 4000 images and 40 different styles. The range of values for each metric is: *ARI*: -1 to 1, *NMI*: 0 to 1, *SC*: -1 to 1, and *CHI*: 0 to $\infty$. The **Base** column indicates the SC and CHI values with perfect ground truth and no modification to the input embedding. We observe a similar trend with the MMC dataset as the MSC dataset as both datasets are created using style-transfer methods. *As *Mamba-ST* was used to create the curated dataset, we excluded $F_{Mamba}$ from ranking.

| Features | | ARI | | NMI | | SC | | | CHI | | |
|---|---|---|---|---|---|---|---|---|---|---|---|
| | | K-Means | DEC | K-Means | DEC | Base | K-Means | DEC | Base | K-Means | DEC |
| $F_{Generic}$ | $F_{Dense}$ | 0.164 | 0.049 | 0.25 | 0.13 | 0.02 | 0.097 | 0.959 | 21.49 | 46.26 | 120196.98 |
| | $F_{LongCLIP}$ | 0.044 | 0.041 | 0.092 | 0.085 | 0.19 | 0.179 | 0.387 | 97.34 | 82.85 | 2347.93 |
| | $F_{DINO}$ | 0.058 | 0.049 | 0.115 | 0.103 | 0.181 | 0.164 | 0.345 | 106.91 | 79.01 | 1923.11 |
| $F_{StyleFeat}$ | $F_{Gram}$ | 0.816 | 0.698 | 0.961 | 0.926 | 0.29 | 0.259 | 0.514 | 613.29 | 618.8 | 3750.41 |
| | $F_{g.c}$ | 0.134 | 0.129 | 0.448 | 0.446 | -0.096 | 0.16 | 0.52 | 2205.62 | 9985.58 | 89669.42 |
| | $F_{IntroStyle}$ | 0.28 | 0.241 | 0.41 | 0.391 | 0.085 | 0.08 | 0.235 | 109.31 | 87 | 1233.49 |
| $F_{ST}$ | $F_{StyleGAN}$ | 0.417 | 0.384 | 0.669 | 0.634 | -0.03 | -0.005 | 0.909 | 17.43 | 18.82 | 19549.91 |
| | $F_{Stytr2}$ | 0.98 | 0.991 | 0.99 | 0.995 | 0.6 | 0.6 | 0.719 | 4368.14 | 4476 | 15056.43 |
| | *$F_{Mamba}$ | 0.9 | 0.836 | 0.97 | 0.94 | 0.468 | 0.45 | 0.608 | 617.04 | 617.93 | 7992.44 |
| | $F_{Styleshot}$ | 0.96 | 0.748 | 0.98 | 0.938 | 0.304 | 0.28 | 0.631 | 304.53 | 298.51 | 5845.3 |
| | $F_{DEADiff}$ | 0.039 | 0.028 | 0.09 | 0.073 | 0.123 | 0.11 | 0.463 | 53.47 | 41.93 | 56413.32 |
| $F_{Lang}$ | $F_{StyleCap}$ | 0.03 | 0.01 | 0.092 | 0.068 | -0.02 | 0.096 | 0.963 | 8.027 | 48.83 | 178514.62 |
| | $F_{Annot}$ | 0.053 | 0.051 | 0.21 | 0.212 | -0.037 | 0.024 | 0.936 | 15.37 | 44.64 | 38490.53 |
| $F_{Train}$ | $F_{CSD}$ | 0.86 | 0.632 | 0.94 | 0.839 | 0.141 | 0.13 | 0.299 | 101.6 | 100.24 | 1413.48 |
| | $F_{Artist}$ | 0.00004 | 0.00002 | 0.05 | 0.04 | 0.091 | 0.07 | 0.157 | 67.31 | 52.59 | 139.42 |
| | $F_{ArtMove}$ | 0.0001 | 0.0001 | 0.06 | 0.04 | 0.103 | 0.08 | 0.171 | 71.56 | 59.42 | 125.71 |

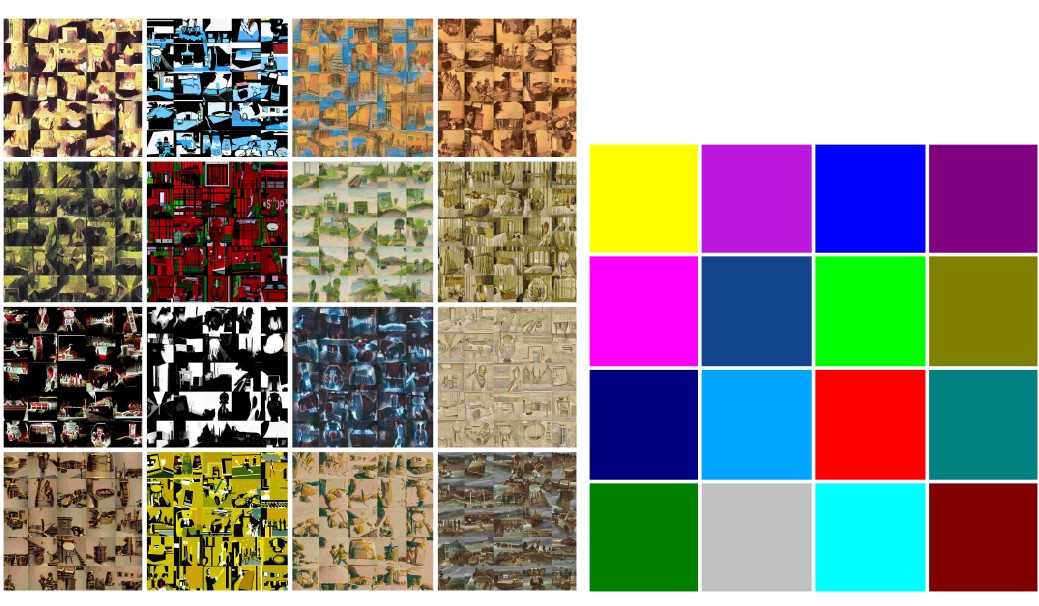

(a) *Ground Truth* style clusters      (b) Class distribution in *Ground Truth* clusters

Figure 24: Artwork samples from the *Synthetically Curated StyleShot* dataset serving as the ground truth for qualitative comparison of clustering with different neural representations.

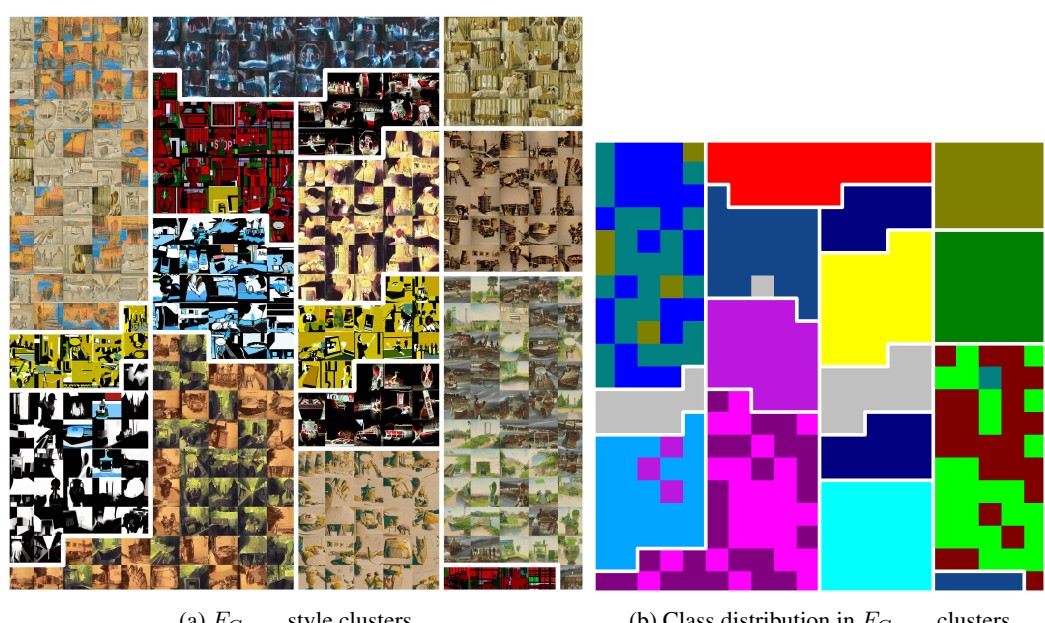

(a) $F_{Gram}$ style clusters

(b) Class distribution in $F_{Gram}$ clusters

Figure 25: Qualitative results of style clustering on the sample *Synthetically Curated StyleShot* dataset (Fig. 24) using $F_{Gram}$ (category: $F_{StyleFeat}$) neural style representation.

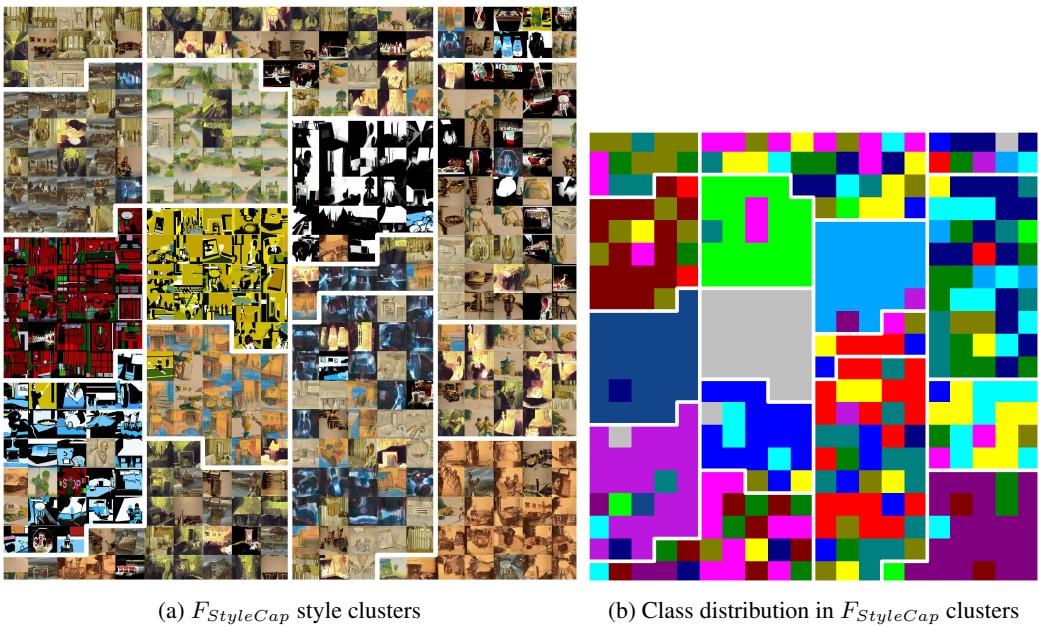

(a) $F_{StyleCap}$ style clusters

(b) Class distribution in $F_{StyleCap}$ clusters

Figure 26: Qualitative results of style clustering on the sample *Synthetically Curated StyleShot* dataset (Fig. 24) using $F_{StyleCap}$ (category: $F_{Lang}$) neural style representation.

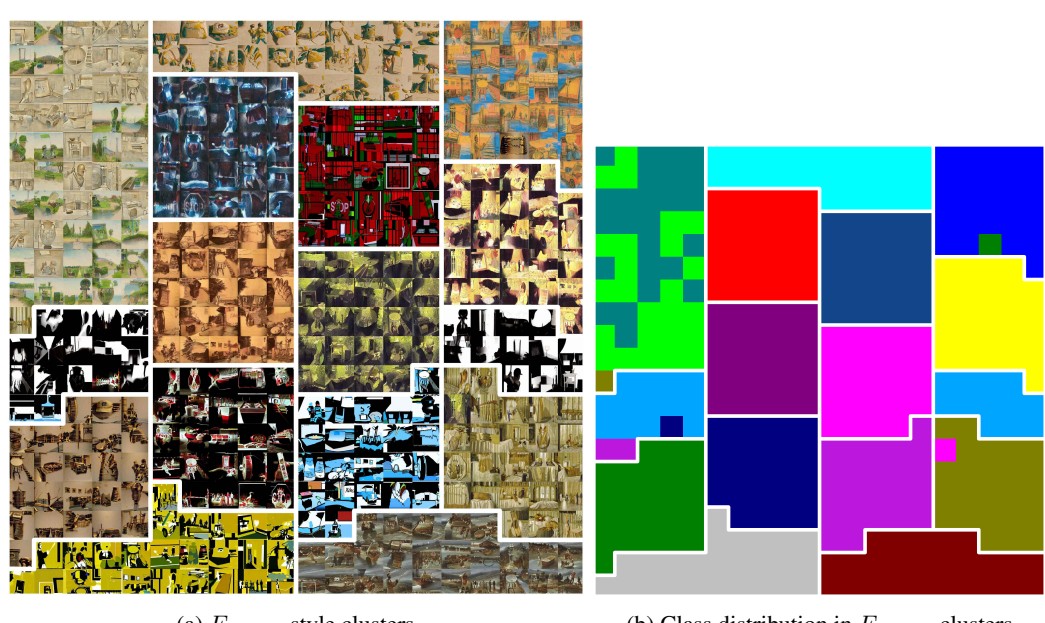

(a) $F_{Stytr2}$ style clusters

(b) Class distribution in $F_{Stytr2}$ clusters

Figure 27: Qualitative results of style clustering on the sample *Synthetically Curated StyleShot* dataset (Fig. 24) using $F_{Stytr2}$ (category: $F_{ST}$) neural style representation.

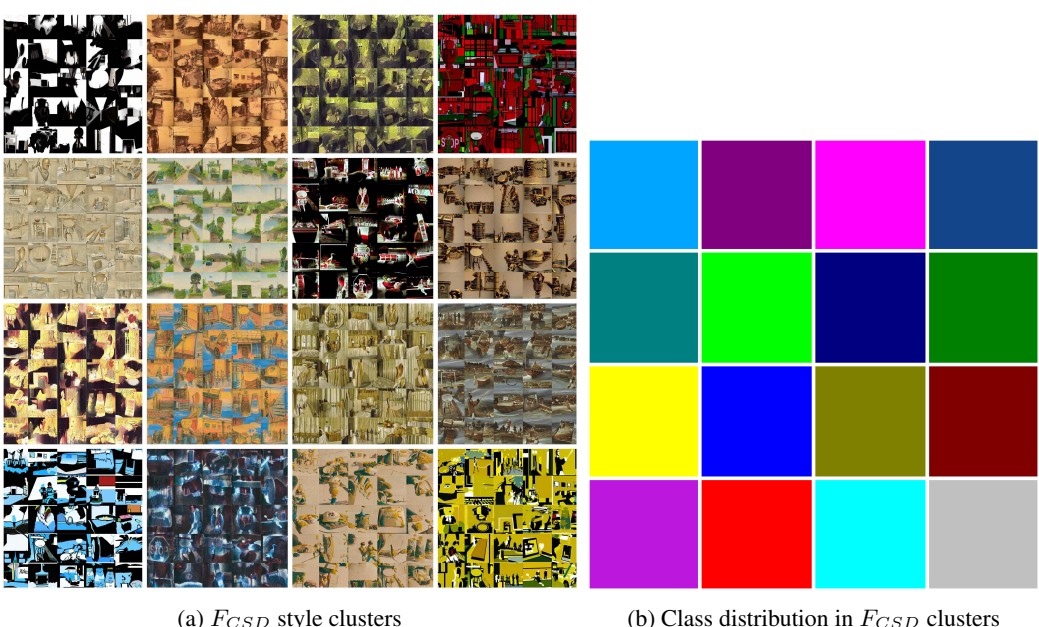

(a) $F_{CSD}$ style clusters

(b) Class distribution in $F_{CSD}$ clusters

Figure 28: Qualitative results of style clustering on the sample *Synthetically Curated StyleShot* dataset (Fig. 24) using $F_{CSD}$ (category: $F_{Train}$) neural style representation.

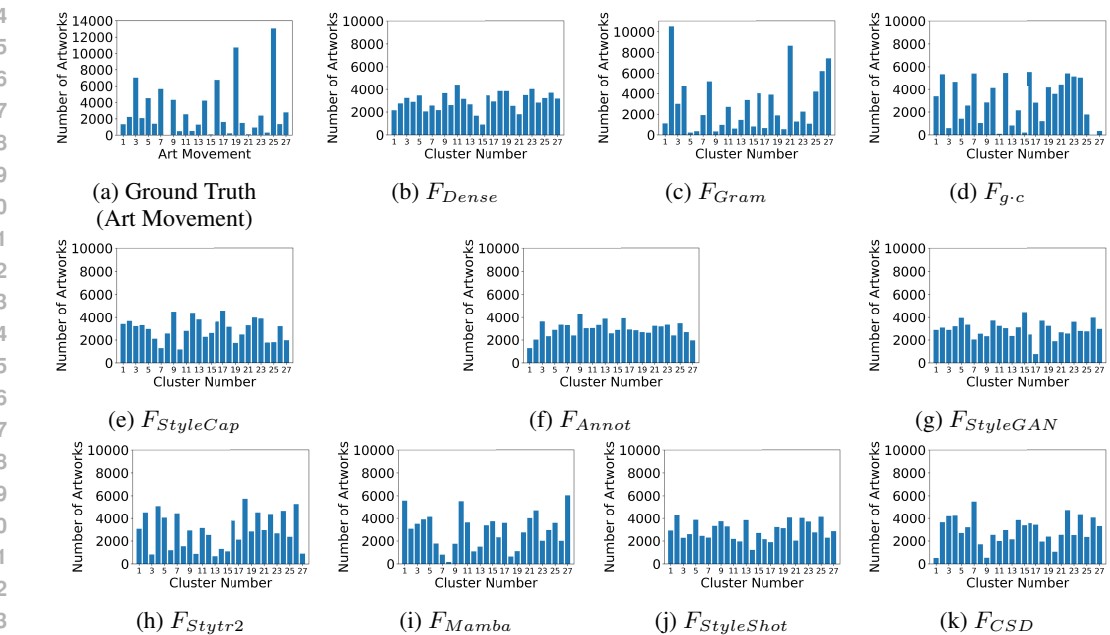

(a) Ground Truth
(Art Movement)

(b) $F_{Dense}$

(c) $F_{Gram}$

(d) $F_{g \cdot c}$

(e) $F_{StyleCap}$

(f) $F_{Annot}$

(g) $F_{StyleGAN}$

(h) $F_{Stytr2}$

(i) $F_{Mamba}$

(j) $F_{StyleShot}$

(k) $F_{CSD}$

Figure 29: Cluster distributions for each representation obtained on the Wikiart-AM dataset.

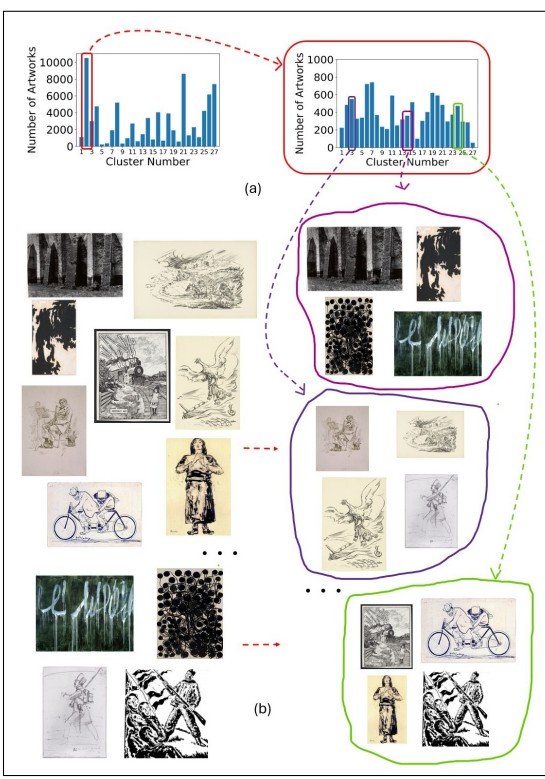

Figure 30: Sub-clustering on a single cluster from the results of the WikiArt-AM dataset for $F_{Gram}$ features through the DEC model. (a) shows the distribution of the number of samples in each cluster before and after sub-clustering. (b) shows the qualitative results after we obtain the sub clusters of a single cluster with most samples. Samples on the left are from the original cluster and samples on the right are from the sub-clusters.

# H HIERARCHICAL NATURE OF STYLE RELATIONSHIPS IN ARTWORK DATASETS

Style-based clustering reveals fundamental insights into artistic datasets, particularly the hierarchical organization of stylistic relationships. Our experiments demonstrate that styles in artwork datasets exhibit inherent hierarchical structures, supported by the following empirical evidence:

## H.1 UNEVEN CLUSTER DISTRIBUTION INDICATES HIERARCHICAL ORGANIZATION

Analysis of cluster distributions across different representations (Figure 29) on the WikiArt dataset reveals significant imbalances in cluster assignments. Representations such as $F_{Gram}$ and $F_{g \cdot c}$ exhibit pronounced peaks, with the majority of artworks concentrated in only 1-3 clusters, creating highly uneven cluster sizes. An initial examination of these dominant clusters shows artworks that appear stylistically similar. However, sub-clustering analysis reveals finer-grained stylistic distinctions.

For instance, when we apply sub-clustering to the largest cluster in the WikiArt-AM dataset using the $F_{Gram}$ representation (Figure 30), the seemingly homogeneous style cluster decomposes into multiple distinct stylistic subclusters. This decomposition demonstrates that high correlations within representations, such as $F_{Gram}$, initially mask the underlying stylistic diversity, confirming the presence of hierarchical style structures.

## H.2 HIERARCHICAL CLUSTERING REVEALS MULTI-LEVEL STYLE ORGANIZATION

We applied hierarchical clustering to the WikiArt dataset using $F_{StyleCAP}$ representations for both art movement and artist-based style definitions (Figures 31 and 32). The resulting dendrograms clearly illustrate hierarchical style relationships at multiple granularity levels.

### H.2.1 ART MOVEMENT HIERARCHY

At the highest hierarchical level, art movements with similar stylistic foundations cluster together. For example, Minimalism and Color Field Painting initially group due to their shared abstract characteristics. As we traverse down the hierarchy, these movements separate into distinct clusters, reflecting nuanced differences in their abstract styles. Similarly, Pointillism, High Renaissance, and Northern Renaissance initially cluster together based on their common focus on realistic subject matter. At deeper hierarchical levels, Pointillism separates first due to its distinctive pointillist technique, while High Renaissance and Northern Renaissance eventually split, revealing subtle stylistic differences between these Renaissance movements.

### H.2.2 ARTIST-LEVEL HIERARCHY

The hierarchical structure extends to individual artists (Figure 32). At the top level, abstract and cubist artists—Pablo Picasso, Juan Gris, and Georges Braque—form one cluster, while realist and symbolist artists like John French Sloan and Nicholas Roerich form another, clearly distinguishing abstract from representational styles. Deeper in the hierarchy, even artists within the same movement separate into distinct clusters. For instance, despite both being prominent cubist artists, Juan Gris and Pablo Picasso ultimately occupy different clusters, reflecting their individual interpretations and techniques within the cubist movement.

These hierarchical patterns demonstrate that artistic styles exist at multiple levels of abstraction, from broad categorical distinctions to subtle individual variations, providing a rich framework for understanding stylistic relationships in artwork datasets.

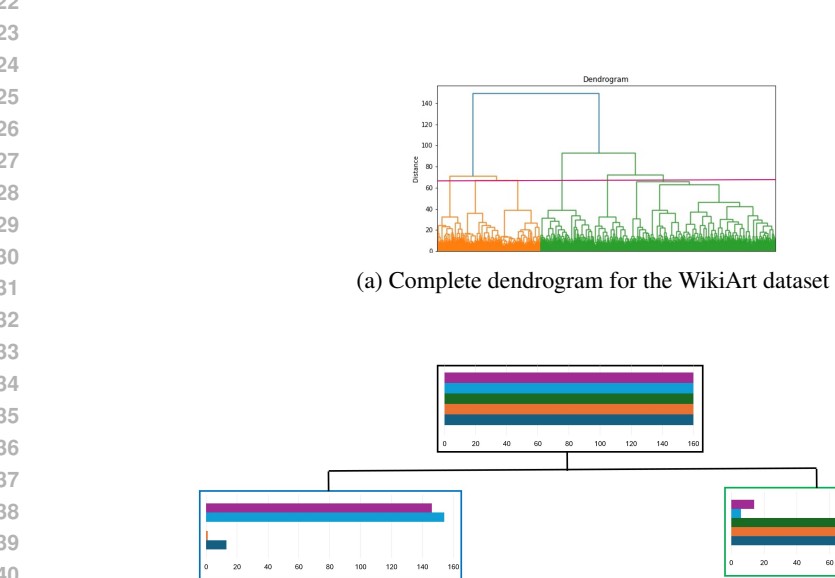

(a) Complete dendrogram for the WikiArt dataset

(b) Sample Art Movement distribution dendrogram

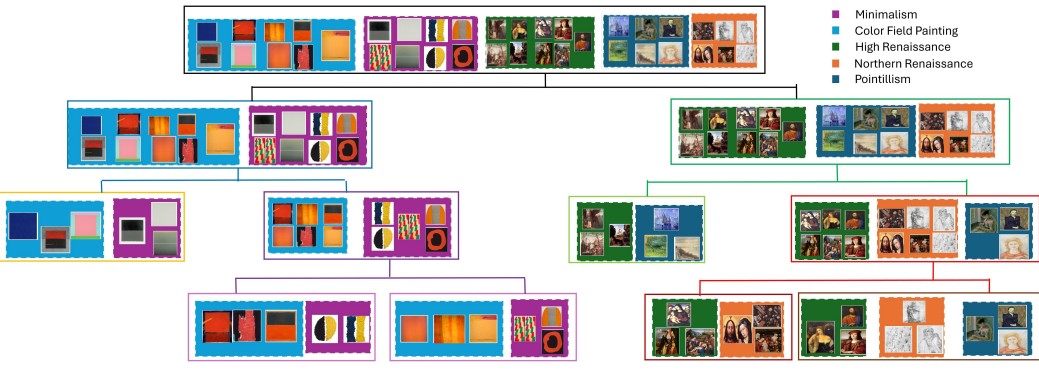

(c) Sample Artworks from each level of the hierarchy based on *Art Movement* categorization

Figure 31: Hierarchical distribution of *Art Movements* in the *WikiArt dataset*. We showcase the sample art movement-wise artworks distribution dendrogram in (b) and the respective sample artworks in (c). The dendrogram is obtained with 27 art movements with the $F_{StyleCap}$ features. We display the top 5 art movements. We observe that the WikiArt dataset contains hierarchies showcasing a higher level of similarity between art movements at the top of the hierarchy. The art movements get separated into distinct clusters when we move down the hierarchy.

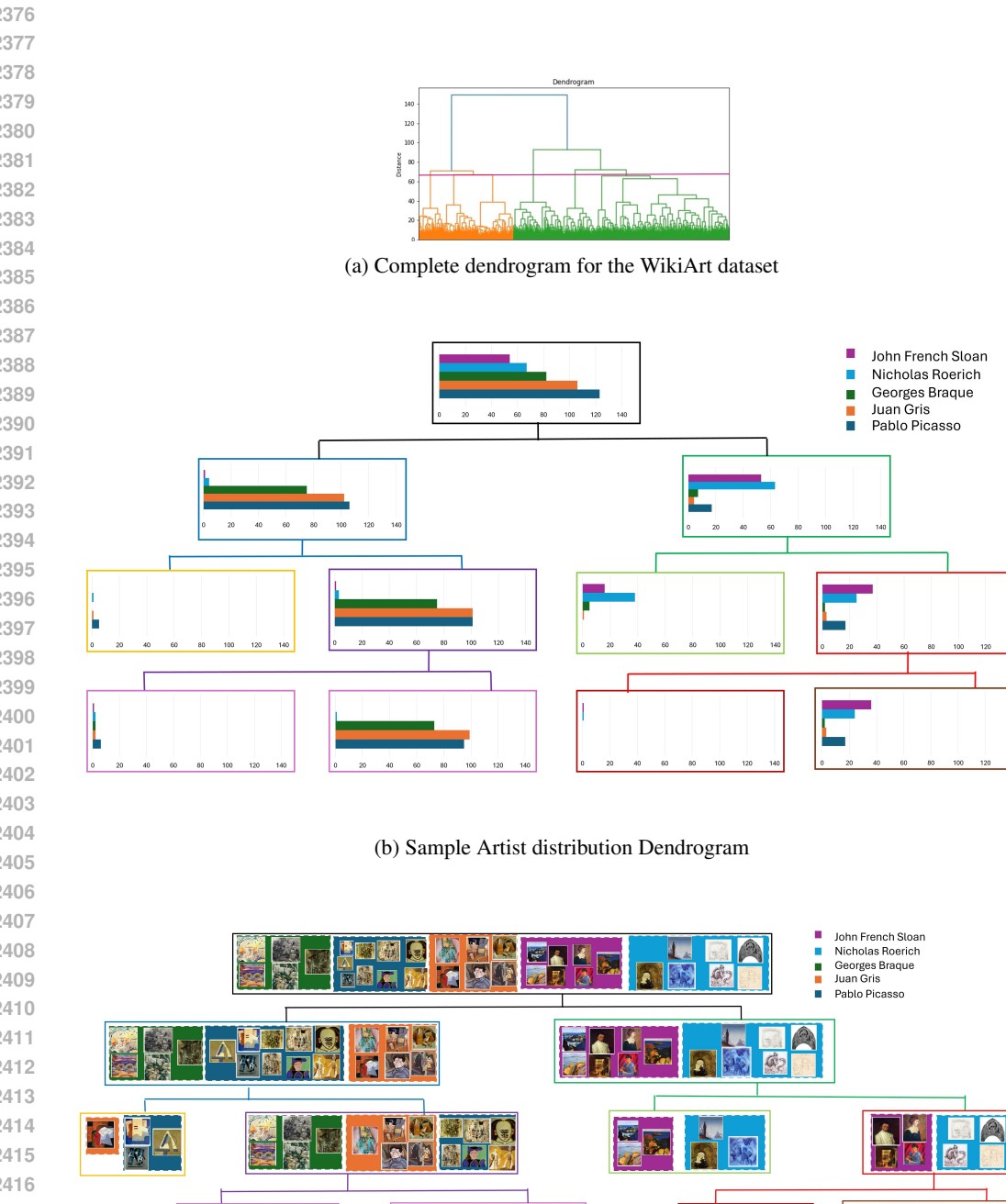

(a) Complete dendrogram for the WikiArt dataset

(b) Sample Artist distribution Dendrogram

(c) Sample Artworks from each level of the hierarchy based on *Artist* categorization

Figure 32: Hierarchical distribution of *Artists* in the *WikiArt dataset*. We showcase the sample artist-wise distribution dendrogram in (b) and the respective sample artworks in (c). The dendrogram is obtained with 765 artists with the $F_{StyleCap}$ features. We display the top 5 artists in this dataset based on the number of artworks. We observe a hierarchical trend similar to the *art movements* distribution. The artists with similar styles are grouped at the top of the hierarchy, whereas the artists get separated into different clusters when we move down the hierarchy.

