# OpenReview forum: "Analyzing Neural Style Representations for Unsupervised Clustering: Visual Art as a Testbed"
_ICLR.cc/2026/Conference — ICLR 2026 Conference Withdrawn Submission_

### Official Review · Reviewer_8ivT · 2025-10-27

**Soundness:** 2
**Presentation:** 3
**Contribution:** 2
**Rating:** 4
**Confidence:** 4

**Summary:**

This paper delivers a large-scale, systematic benchmark of neural style representations for unsupervised clustering in visual art. It evaluates 16 representation families (task-based, style-transfer, and language-based, including a few novel variants) with two clustering paradigms across real and synthetic datasets that encode different operationalizations of “style.” Metrics include ARI, NMI, Silhouette, and Calinski–Harabasz, complemented by a small human perceptual study. The central finding is clear: specialized style features generally outperform generic ones on perceptual cohesion, yet no single representation works across all definitions of style—highlighting the concept’s inherent ambiguity and positioning visual art as a challenging testbed for unsupervised representation learning.

**Strengths:**

This paper demonstrates exceptional rigor through a comprehensive benchmark spanning 16 representation families and multiple datasets, unified by a carefully designed evaluation protocol that respects competing definitions of "style." The authors employ thoughtfully constructed synthetic style-transfer datasets to probe style-content disentanglement in ways that real annotations cannot easily support, while incorporating a perceptual study that strengthens the finding that "ground truth" style is inherently subjective and provides valuable context for interpreting metric-only results. Throughout the work, claims are consistently grounded in quantitative outcomes, and the visualizations effectively illustrate the meaningful gap between art-historical labels and perceptual clusters, making the evidence both compelling and accessible.

**Weaknesses:**

The paper suffers from significant under-specification of methods and notation that undermines reproducibility. Key implementation details remain missing or inconsistent throughout, including exact formulations for Gram-based and language-derived features, layer selection strategies, normalization and whitening procedures, dimensionality reduction pipelines, and clustering configurations. Metric formulas are incompletely documented, and the handling of edge cases such as class imbalance and degenerate partitions is not adequately addressed, leaving readers unable to faithfully replicate the experimental setup.

The choice of clustering baselines is notably narrow, limiting the evaluation to only K-Means and DEC, which weakens the core benchmarking claim. The authors overlook stronger and more directly relevant methods such as Invariant Information Clustering (IIC), spectral and graph-based clustering approaches, and modern contrastive clustering techniques. Inclusion of these methods could meaningfully alter the conclusions about which representations best capture artistic style, and their absence represents a missed opportunity to establish more robust comparative insights.

The human perceptual study, while valuable in principle, is reported with insufficient detail. Critical information about participant recruitment, task instructions, randomization procedures, and inter-rater reliability measures such as Cohen's kappa are not provided. The analysis lacks fine-grained statistics and uncertainty quantification, making it difficult to assess the robustness and generalizability of the human-algorithm alignment findings.

Finally, the paper demonstrates limited analytical depth and notable gaps in the literature review. Concrete failure cases are scarce, hyperparameter sensitivity is minimally explored, and important issues such as dataset bias and the synthetic-to-real domain gap receive insufficient discussion. The related work section under-cites or omits several foundational and recent contributions, including Ji et al. (2019) on IIC, Karayev et al. (2013) on recognizing image style, Crowley & Zisserman (2014) on CNN-based painting retrieval, Gonthier et al. (2018) on weakly supervised detection in art, generative art models like ArtGAN (Tan et al., 2018) and CAN (Elgammal et al., 2017), and more recent art-focused clustering work such as Gultepe et al. (2025) and Hu et al. (2023).

**Questions:**

Could you provide complete, reproducible implementation details and formulas for all representation families evaluated?

Why were stronger unsupervised clustering baselines not included in the benchmark？

The human perceptual study requires substantially more methodological transparency. Can you provide detailed information on participant recruitment procedures, exact task instructions, randomization protocols, and inter-rater reliability statistics such as Cohen's kappa? Furthermore, please include finer-grained statistical analysis with confidence intervals or error bars, and offer a systematic correspondence analysis between human judgments and each algorithmic metric to strengthen the human-algorithm alignment claims.

---

### Official Review · Reviewer_AdRZ · 2025-10-28

**Soundness:** 3
**Presentation:** 3
**Contribution:** 2
**Rating:** 2
**Confidence:** 4

**Summary:**

This paper presents a comprehensive analysis of multiple art datasets, using different definitions of style, and two different clustering algorithms (no similarity measures considered or mentioned). The major conclusion is that unsupervised clustering for art to find style is difficult and hence we can conclude that the information that is needed is not sufficiently captured in the low-level embeddings.

**Strengths:**

- The paper gives an elaborate analysis including a large appendix where additional results are presented.
- Good overview of the role of different aspects of style in art and how that is / is not reflected in the clustering results.
- The paper is well written and easy to follow with many figures that give a good overview of the results. Findings are also presented in a well structured manner.

**Weaknesses:**

- There is only an indication that this could also be applied to other domains. For now the paper is only relevant for art so probably is more suited for publication in a more digital humanities venue. Also reflected in the abstract where the contributions mentioned are digital curation, cultural heritage, and style-aware computer vision. The latter is only discussed in a very limited way and not backed by evidence beyond art. Furthermore, the references are not including ICLR.
- The paper considers one method by the authors but that method is not outperforming the rest and also doesn't yield a clear complementary perspective.
- The procedure is a well designed but standard experimental protocol. Not sufficiently innovative to be a blueprint for other application domains.
- The clustering techniques are rather old. More modern methods should be considered.

MINOR:
- Many references are incomplete.

**Questions:**

- What perspective does you own method bring that is not already covered by the other methods?
- How would this method generalize to other domains?

**Details Of Ethics Concerns:**

N.A.

---

### Official Review · Reviewer_RF1M · 2025-10-30

**Soundness:** 3
**Presentation:** 3
**Contribution:** 2
**Rating:** 4
**Confidence:** 4

**Summary:**

This paper focuses on the inability of neural style representations to effectively support unsupervised clustering of visual artworks, despite their widespread use in style classification and transfer. It systematically evaluates 16 state-of-the-art neural style representations across 4 datasets (covering diverse style definitions) and 2 clustering algorithms, aiming to clarify how these representations capture "style"—a concept with inherent ambiguity (art movements, artist signatures, perceptual attributes, etc.). The paper develops a unified framework integrating datasets, clustering algorithms, and metrics to compare style representations. To address the limitation of real-world datasets (where style and content are often entangled), the paper introduces two synthetic curated datasets (MSC/MMC). Through cluster distribution analysis and hierarchical dendrograms, the paper demonstrates that artistic styles exhibit inherent multi-level relationships.

**Strengths:**

The paper demonstrates strong originality by addressing longstanding limitations in style representation research and introducing creative solutions that expand the scope of unsupervised clustering. It maintains exceptional methodological quality through well-controlled experiments, multi-faceted validation, and detailed documentation—ensuring reproducibility and credibility. The paper is exceptionally clear, making complex technical concepts and experimental results accessible to readers across computer vision, art informatics, and machine learning.

**Weaknesses:**

The key results that the study yields three main insights seems trivial and with restricted specific value to guide the representation learning investigation in the future.

**Questions:**

Thank you for conducting extensive experiments and presenting several insightful new findings in the paper. Could the authors explain the specific guiding significance of these new insights for future research work?

---

### Official Review · Reviewer_Jmjt · 2025-11-01

**Soundness:** 3
**Presentation:** 3
**Contribution:** 3
**Rating:** 6
**Confidence:** 3

**Summary:**

This paper claims to be the first systematic benchmark for unsupervised style clustering of artworks: it compares 16 neural representations with K-Means and DEC across four datasets that operationalize distinct style notions—WikiArt–ArtMove (27 movements), WikiArt–Artist (40 artists), DomainNet-3k (6 styles × 50 contents), and two synthetic curated sets (MSC/MMC) introduced here via style transfer for controlled style–content disentanglement. Evaluation combines external (ARI, NMI) and internal (Silhouette, Calinski–Harabasz) metrics plus a 25-participant perceptual study. The contributions of this study include language-based style features (FStyleCap, FAnnot) and artwork-trained Vision Transformers (FArtMove, FArtist). DEC improves internal geometry without altering external alignment, indicating that cleaner clusters do not always align better with ground-truth style categories.

**Strengths:**

1. The motivation is clear: style has multiple meanings, and there are few available labels; therefore, unsupervised analysis serves as an effective stress test. The paper effectively addresses this gap.

2. Broad coverage. 16 embeddings spanning CNN/ViT, style transfer, diffusion, VLMs, and the two new language features, all under one protocol. Method families are laid out cleanly.

3. Multi-view datasets. Beyond WikiArt and DomainNet-3k, the new MSC/MMC synthetic sets enable controlled checks of style–content disentanglement.

4. Insightful results. On WikiArt-Artist, FCSD and FArtist beat generic features; on synthetic data, style-transfer family features (e.g., StyleShot/Mamba-ST derived) dominate—evidence that no single embedding fits all “style” definitions.
5. Two-axis evaluation. Internal vs. external metrics are separated, and the DEC analysis makes the geometry-vs-semantics tension explicit.
6. Perceptual study. Human ratings show that perceived style can diverge from art-historical labels—useful nuance for this area.

**Weaknesses:**

1. Mostly point estimates; variance across seeds/initializations and significance tests are missing for K-Means/DEC.
2. Several settings appear to set K = number of ground-truth classes (e.g., DomainNet-3k), which injects label priors into an “unsupervised” evaluation.
3. MSC/MMC are built with specific style-transfer engines; even with “exclude same-source features,” rankings may reflect generator artifacts.
4. FStyleCap/FAnnot often show higher internal but modest external scores; the claim that they capture interpretable style cues would benefit from direct analysis.
5. No hierarchical/spectral/density methods despite hints of hierarchical style structure.

**Questions:**

1. Can you add an unsupervised K selection (e.g., Silhouette/CHI/information criteria) and report how method rankings and key claims change relative to K = ground-truth?
2. For K-Means/DEC, report mean ± std over multiple seeds, significance tests, and DEC stopping/temperature details. Do any conclusions flip under different inits?
3. Rebuild MSC/MMC with different style-transfer engines and disjoint style pools (OOD); redo evaluation and report rank stability. Which methods are most/least sensitive?
4. Given the paper’s claim of hierarchical/structured style, why limit to K-Means/DEC? Can you include hierarchical, spectral, and density-based (e.g., HDBSCAN) baselines and test whether the finding holds?

---

### Note · Authors · 2025-12-08

**Comment:**

We regret to inform that we are withdrawing our paper. We believe the present conference format and page restrictions do not allow us to fully convey our ideas. Many of the questions raised by the reviewers could have been easily addressed in a more comprehensive exposition. This has led to our work being misunderstood or not viewed in the intended perspective.

We are exploring alternative formats that better align with the intent of this paper. We sincerely thank the reviewers for their valuable feedback and remarks, some of which we greatly appreciate and will incorporate to improve our future work.

**Withdrawal Confirmation:**

I have read and agree with the venue's withdrawal policy on behalf of myself and my co-authors.